# Safeguarding against external intrusions utilizing adaptive bio-inspired multi-population anomaly detection for IoT network

**Shubhra Dwivedi[1], Alok Kumar Shukla[1], Diwakar Tripathi[2], Sunil Kumar Singh [ID][3]***

**1** Thapar Institute of Engineering and Technology, Patiala, Punjab, India, **2** National Institute of Technology Jamshedpur, Jamshedpur, India, **3** Manipal Institute of Technology Bengaluru, Manipal Academy of Higher Education, Manipal, India

* sunil.sisngh@manipal.edu

## Abstract

The rapid growth of Internet of Things (IoT) devices has dramatically increased demand for robust, adaptive security solutions capable of countering the growing sophistication of cyberattacks. Despite extensive research efforts focused on anomaly-based intrusion detection systems tailored to IoT network traffic, conventional detection frameworks often fail to effectively identify novel or zero-day attack patterns, thereby falling short of the dynamic security requirements of modern IoT ecosystems. To address these critical limitations, this study introduces a novel anomaly-based intrusion detection system called Chaotic Multi-Population Grasshopper Optimization with Differential Evolution (CMGODE). The proposed approach significantly enhances the standard Grasshopper Optimization Algorithm by integrating chaotic mapping mechanisms to improve exploitation and prevent premature convergence, adopting a multi-population strategy to maintain diversity and enhance global search, and incorporating a differential evolution-based refinement phase to improve the quality of global candidate solutions. The effectiveness of the CMGODE-based detection system is thoroughly evaluated on two widely adopted benchmark datasets, namely BoT-IoT and UNSW-NB15. Experimental results demonstrated that our proposed method achieved an excellent balance between high detection accuracy and computational efficiency, consistently outperforming several state-of-the-art approaches in accurately identifying both known and previously unseen attacks within IoT network environments.

## 1. Introduction

The Internet of Things (IoT) is widely regarded as one of the most rapidly evolving technologies nowadays, driven by the explosive expansion of ubiquitous connectivity and the increasingly seamless integration between the physical and virtual worlds [1]. The direct bridge between the physical and digital realms created by

**Data availability statement:** The data underlying the results presented in the study are available from Kaggle. The UNSW-NB15 dataset is publicly available from: https://doi.org/10.34740/kaggle/dsv/9350725, and the Bot-IoT dataset is publicly available from: https://www.kaggle.com/datasets/vigneshvenkateswaran/bot-iot.

**Funding:** Manipal Institute of Technology Bengaluru, MAHE Bengaluru Campus the funder had a role in study design and manuscript preparation.

**Competing interests:** NO authors have competing interests.

the Internet of Things has raised significant concerns about targeted cyber-attacks on these devices, necessitating robust security measures. This critical vulnerability is shown by high-consequence incidents in which compromised IoT devices have caused substantial real-world damage and disruption. Notable cases include the 2015 cyber-attack on Ukraine's power grid, which caused widespread blackouts; the successful cyber-induced shutdown of a floating oil rig by tilting its control systems; and the co-opting of smart refrigerators and other appliances into massive botnets used for malicious activities like spam distribution. These events collectively underscore the tangible and severe risks posed by inadequate IoT security, moving beyond data theft to enable direct physical and infrastructural harm [2]. Additionally, one particularly infamous attack in the history of IoT occurred in 2017, when the Mirai malware infected over 380,000 IoT devices, creating widespread botnets to distribute Denial-of-Service (DoS) attacks. This malware remains the most significant recorded DDoS attack, according to the US-CERT [3].

To mitigate threats to IoT devices and support robust security mechanisms, intrusion detection systems (IDSs) serve as a critical defence layer [4]. These systems continuously monitor network traffic and device behaviour to detect and respond to a wide range of security threats. In general, an intrusion detection system is a software or hardware system designed to monitor for signs of malicious cyber-attacks in computer networks [5]. According to past literature, IDS is considered into signature-based and anomaly-based approaches [6]. Building upon reputable taxonomy, signature-based detection, also known as misuse-based detection, offers high accuracy for known threats but fails against novel, zero-day attacks for which no signature exists. Conversely, anomaly-based detection addresses this gap by modelling normal behaviour to flag deviations, proving effective against new attack vectors but often at the cost of higher false-positive rates. The hybrid approach synthesises these methods, leveraging the precision of signature matching for known threats while employing anomaly detection to identify suspicious novel activities, making it particularly suited for the dynamic, heterogeneous, and complex environments characteristic of modern IoT networks [7].

The inherent limitations of IoT data, characterised by simple feature sets and a scarcity of knowledge-based attributes, present a dual challenge for security. These constraints not only complicate the fundamental task of data protection but also significantly impede the performance of intrusion detection systems. The resulting models often exhibit reduced detection rates, and processing such data demands substantial computational time and resources, creating a substantial operational bottleneck [8]. As a result, selecting the suitable features to build a mechanism is crucial for increasing speed, reducing costs, and improving comprehension of test results. To overcome the aforementioned limitations, feature selection (FS) plays a vital role in addressing key challenges in intrusion detection systems by identifying the most relevant characteristics for effective resolve cybersecurity concerns [9]. Specifically, FS algorithms are naturally divided into three categories such as filter, wrapper, and hybrid techniques [10]. Compared to the filter methods, the wrapper approaches

surpasses it, although it requires more processing time [11]. At the same time, hybrid feature selection methods combine the strengths of both filter and wrapper approaches [12].

In the domain of cybersecurity, particularly within anomaly-based intrusion detection systems (IDSs), a primary challenge lies in accurately distinguishing malicious attacks from the vast and highly diverse spectrum of normal network behaviour. This difficulty is often exacerbated by limited model accuracy and the widespread presence of high-dimensional data containing numerous irrelevant or redundant features. The curse of dimensionality further complicates matters, as increased feature dimensions lead to data scarcity, keen noise levels, and reduced effectiveness of traditional detection techniques, ultimately resulting in elevated false positive rates, challenges in establishing reliable normal baselines in dynamic environments. To address the limitations of existing feature selection methods, wrapper-based techniques have proven particularly effective by directly evaluating feature subsets according to their impact on the actual performance of the intrusion detection system (IDS) model. Among wrapper approaches, evolutionary algorithms represent one of the most prominent and powerful population-based strategies, thanks to their ability to efficiently explore large, complex search spaces. However, optimization performance of these algorithms heavily depends on the proper tuning of their standard control parameters including population size, crossover rate, and mutation rate [13]. In this regard, evolutionary algorithms such as the genetic algorithm, particle swarm optimisation, teaching learning-based optimisation, differential evolution, artificial bee colony, and grasshopper optimisation algorithm are receiving increasing attention in evolutionary research [14]. However, incorrect tuning of these factors results in a tendency towards a locally optimal solution or increased computational cost. Therefore, researchers are drawn to this field of study because of the appropriate dataset dimension and anomaly detection in computational intelligence [15].

Previous literatures have introduced several EA-based methods for intrusion detection that achieved high performance [16]. Despite their accuracy in detecting attacks, these methods required extensive analysis time and had high computational costs during training. Moreover, basic GOA technique still suffers from several inadequacies, including poor convergence and the risk of getting stuck in local optima. However, the continuous expansion of Evolutionary Algorithms (EAs) and the increasing complexity of data have led to more complex and diverse intrusions, posing severe challenges for IoT security. To address these issues, this study focuses on a new algorithm, CMGODE that combines chaotic multi-population GOA with the DE strategy to select significant features for monitoring malicious traffic in IoT networks. Firstly, chaos mechanism is integrated into GOA to explore other search areas before the population updating process in CGOA. Then, in CGOA, from global search to precise area search, populations are divided into sub-populations using multi-population strategy to explore the global space more deeply. The iterative process gradually shifts, and the updated populations are further improved using the DE strategy. The DE scheme utilises three operators to generate more effective agents. Finally, three SVM classifiers are used to evaluate solution fitness, and the best is then used for further evaluation. The main contributions of this paper are as follows:

- Firstly, the chaotic mechanism introduces a trade-off between exploration and exploitation, thereby balancing the basic GOA more effectively.

- Secondly, the basic GOA method divides all populations into sub-populations according to the multi-population strategy required to explore the global space.

- Lastly, new populations are processed by three DE operators and integrated into GOA to enhance the detection performance for anomaly-based network intrusion detection, while also reducing the computational time and false alarm rate.

- The proposed method is studied to measure the effectiveness and timely detection of malicious traffic in the IoT network environment on BoT-IoT and UNSW-NB15. According to the outcomes, the primary advantage of early detection of malicious traffic in the IoT network is the prevention of significant exploitation and the isolation of IoT devices from malicious nodes.

   

## 1.1. Novelty of our method

The fundamental objective of this research is to enhance the performance of anomaly-based intrusion detection systems by developing and applying a range of Evolutionary Algorithm (EA)-based learning methods. While preliminary results indicate a reasonable level of improvement, the achieved performance suggests there remains a significant opportunity for further optimisation and refinement of the EA techniques [17]. Specifically, these approaches are frequently characterised by high false-positive rates, primarily because they either entirely or partially ignore the critical correlations between features. Consequently, developing an accurate model of normal behaviour becomes inherently difficult, further exacerbating the rate of erroneous alerts. This persistent issue not only drastically limits the system practical detection efficacy but also significantly increases the analytical burden on security researchers, who can expend considerable effort to triage and validate numerous false alarms [18]. Meanwhile, the attack-detection principle holds that the system can automatically improve its performance as it gains experience, thereby mitigating this machine-learning difficulty. In this regard, it is worth noting that, for the first time, we have combined chaotic multi-population GOA with the DE method, known as CMGODE framework, tailored to enhance anomaly intrusion detection in IoT environments. CMGODE, in contrast to prior techniques, integrates principles from chaos theory, multi-population strategies, and evolutionary refinement to enhance both convergence behaviour and overall detection performance. The strategic application of chaotic maps during the initialisation and update phases actively promotes population diversity, thereby preventing premature convergence to suboptimal solutions. Simultaneously, its multi-population co-evolution methodology enables distinct sub-populations to evolve independently while periodically exchanging elite solutions. This framework facilitates a more comprehensive exploration of the search space and effectively mitigates the entrapment in local optima, a well-documented limitation in conventional metaheuristic algorithms.

Furthermore, CMGODE includes Differential Evolution (DE)-based refinement step following the initial search phase, which improves solution quality by exploiting promising regions locally. This hybridization considerably increases detection accuracy while decreasing false positives. The proposed method is designed to be computationally efficient, making it typical well-suited for IoT deployments. Its performance and generalizability are rigorously evaluated on two widely recognized benchmark datasets like BoT-IoT and UNSW-NB15. Compared to existing cutting-edge techniques, CMGODE offers a superior trade-off between detection performance and execution time, making it a viable contender for practical implementation in resource-constrained edge environments.

The rest of the paper is organised as follows. Section 2 discusses the background and previous studies related to our work. The details of the techniques utilised in our model for identifying malicious activity in the IoT environment are illustrated in Section 3. Section 4 presents the experimental results and discusses a framework for intrusion detection. Finally, Section 5 concludes the paper.

## 2. Literature study

The global proliferation of the Internet of Things has accompanied an era defined by billions of interconnected smart devices. This rapid and expansive growth in IoT adoption highlights a critical necessity; design of adaptable architectural frameworks coupled with computationally efficient algorithms is paramount to constructing robust and dependable security solutions able to safeguard these large-scale, heterogeneous, and constantly changing network environments [19]. Indeed, the widespread organisational adoption of IoT to enhance operational facilities represents a foundational shift towards a digital era of self-representing objects. Within this paradigm, paramount research challenge is the real-time and offline processing of voluminous data streams to achieve a critical balance: minimising false positives while maintaining an acceptable threshold for accurate attack detection [20]. This challenge is compounded by inherent security vulnerabilities within IoT intuitive design, which are exacerbated by its explosive growth. For instance, in a smart home ecosystem, the very convenience of extensive remote access also creates a broad attack surface, where adversaries readily exploit unpatched vulnerabilities, weak configurations, and default credentials. Consequently, the broad adoption

of these devices creates a significant and urgent imperative to develop and deploy comprehensive protective measures for IoT networks.

Attacks against CPS communication networks can harm essential resources and infrastructure. Accurate prediction of these assaults minimises their impact on target CPS networks. In [21], the author presented a novel hybrid technique for Intrusion prediction on CPS communication networks. A bio-inspired hyper parameter search technique improved deep neural network structures by focusing on the fundamental hyper parameters. They developed a prediction model using a modified neural network structure and tested its performance on two well-known datasets, CIC-IDS2017 and UNSW-NB15. The extensive experimentation demonstrated that the proposed model accurately predicts various attack types with low error and false-positive rates, outperforming comparable state-of-the-art models. A wireless IoT network requires a network-based IDS that can effectively detect a range of attacks from both internal and external attackers, including injection, flooding, and impersonation attacks [22]. Despite various attacks, existing IDSs still struggle to accurately detect them. To address this, a review of current intrusion detection systems for IoT technologies was conducted in [23], focusing on architecture types. Anomaly-based intrusion detection systems have been deployed to monitor network activity and protect IoT devices from intrusions. Overall, [24] proposed an intelligent IDS to protect IoT devices directly connected to it. It can be deployed on low-cost IoT gateways utilising edge computing to detect cyber threats as close to the data source as possible.

The [25] research proposed a novel methodology for addressing data insecurity using deep networks. The suggested model is a deep learning-based network intrusion detection system that employs a chaotic optimisation technique. The approach was pre-processed with data purification and M-squared normalisation. After pre-processing, the unbalanced datasets were balanced using the Extended Synthetic Sampling technique. Following balancing, kernel-assisted principal component analysis was used to extract features from the dataset. The Chaotic Honey Badger optimisation algorithm selected the ideal characteristics. Gated Attention Dual Long Short-Term Memory classifies attacks once all required features have been extracted. The datasets used in the preceding method were TON-IOT and NSL-KDD. The prototype was evaluated based on the following metrics: accuracy, precision, recall, and F1. The suggested model achieves 99.65% accuracy on the NSL-KDD dataset and 98.76% on the TON-IOT dataset. Thus, the model's accuracy and resilience demonstrate that it outperforms other existing models.

The author addressed the main issues in [26] study by proposing two optimised IDSs that integrate grey wolf optimisation (GWO) with deep learning models. The first system combined a gated recurrent unit with GWO, while the second used a long short-term memory network with GWO. These systems strived to improve feature selection, reduce dimensionality, and increase detection accuracy. The NSL-KDD and UNSW-NB15 datasets, which represent contemporary network settings, were used to assess the author systems. The experimental results showed considerable gains in intrusion detection accuracy and computing efficiency, demonstrating the utility of the DL-GWO strategy for improving network security. On the UNSW-NB15 dataset, the first technique (GRU-GWO-FS) improved anomaly and signature-based detection accuracy to 90% and 79%, respectively, compared to 80% and 77% with all features. The second strategy achieved 93% and 79%, respectively, compared to 82% and 77%. On the NSL-KDD dataset, GRU-GWO-FS boosted anomaly detection accuracy to 94% and 92%, while LSTM-GWO-FS improved it to 94% and 92%, respectively.

The author [27] presented a DC intrusion detection system for IIoT that addresses security issues using deep reinforcement learning. The technique that opens the set recognition problem in intrusion detection as a discrete-time Markov decision process was solved using a deep Q-network (DQN). Meanwhile, the DQN value network was updated with a conditional variation auto encoder. The open-set recognition challenge in intrusion detection was broken into two sub-problems: fine-grained traffic classification and the recognition of unknown attacks. They used DQN to solve the fine-grained traffic classification challenge. Reconstruction error was employed to identify unknown attacks, as it is typically lower for known traffic than for unknown attacks. Experiments on the IIoT data set TON-IoT show that the DC-IDS model outperformed earlier methods in terms of recognising unexpected threats and maintaining model stability.

The social internet of things network traffic processing module generates traffic samples, chooses samples for a classifier and a cybersecurity examiner centre, and returns similarities. In [28], DQN with a heuristic learning network is used to gradually increase its ability to detect malicious traffic. Specifically, reward functions were constructed based on the network chosen actions to punish the behaviour of mistakenly labelling malicious samples and make variable reward functions adapt to diverse execution actions. The LSTM-based DQN maximised the cumulative expected reward to determine the best approach for the heuristic learning network. As a result, DQN-HIDS steadily increased the frequency of its labelling behaviour, reduced resource burden, and improved its ability to label SIoT network traffic. Experiments demonstrated the performance of DQN-HIDS in terms of examination centre workload and queue workload for delayed samples, rewards obtained by DQN-based heuristic learning network, and classifier accuracy. Comparisons were also made with cutting-edge deep learning models and traditional machine learning approaches, highlighting the benefits of DQN-HIDS in utilising fewer samples from the Social Internet of Things network traffic.

According to a previous study [23], researchers have adopted machine learning (ML) algorithms, which have played a significant role in ensuring cybersecurity for the Internet of Things by identifying malicious and intrusive traffic. However, due to imprecise feature selection, ML techniques often misclassify malicious traffic in intelligent IoT networks supporting secure intelligent applications. A new FS algorithm based on CorrACC was suggested in [11] to address this issue. It served as a wrapper technique to filter out irrelevant features and select compelling features for specific ML classifiers, utilising an accuracy metric. First, a bi-objective set was used to select operative features, followed by the novel CorrACC feature selection approach. The proposed approach was tested on the BoT-IoT dataset using four different ML classifiers. Similarly, [29] proposed a new BoT-IoT dataset comprising legitimate and simulated IoT network traffic, along with various types of intrusions. A realistic testbed environment was presented to address the shortcomings of existing datasets, capturing complete network information and accurate labelling, as well as recent and complex attack diversity. Different statistical and machine learning methods were employed to assess the reliability of the BoT-IoT dataset relative to benchmark datasets for forensic purposes. It is necessary to use methods that enable several models to work together and adapt to changing network conditions.

The study referenced as [30] directly addressed the current critical knowledge gap by proposing a novel consensus hybrid ensemble model, an advanced ensemble learning technique specifically engineered for intrusion detection in complex environments. Using a meta-classification technique, the author integrated various model types, including ensemble, linear, and nonlinear approaches, in addition to neural networks and probabilistic models. Consensus voting was used in this setup to align predictions from the different base classifiers. The hybrid model of random forests and decision trees serves as a meta-classifier in a voting classifier. By considering each base classifier's confidence and agreement, they approached improved decision-making. The model transparency prediction was influenced by local and global explanation models, including Shapley additive explanations and local interpretable model-agnostic explanations.

The work extends security considerations to both Wireless Sensor Networks (WSNs) and IoT systems, building upon techniques from [31] by investigating existing security attacks to formulate effective prevention strategies. Specifically, to detect a range of widespread network-based cyber-attacks on IoT devices, the layered intrusion detection system proposed in [30] was designed to execute three primary tasks: classifying each IoT device on the network and profiling its expected behaviour; identifying malicious packets during an active attack; and classifying the specific type of attack. This system was validated within a smart home testbed comprising eight commercial devices. In a complementary manner, a comparative study of home energy management controllers utilising heuristic algorithms was conducted in [31], employing integrated Harmony Search, improved Differential Evolution, and a hybrid harmony search differential evolution approach.

Despite the numerous benefits of evolutionary algorithms in improving IDS performance, certain limitations exist, including increased computational time and the risk of being trapped in local optima [32]. There is a need to improve evolutionary techniques to enhance their searchability for finding optimal solutions. This paper proposed chaotic multi-population

GOA with DE approach to select relevant features for more accurate attack classification in the IoT environment, while assessing optimal parameter values and reducing dataset size. While many researchers have made significant efforts, a robust, efficient intrusion detection technique remains necessary to address real-time challenges. This approach utilises the capabilities of a multi-population evolutionary algorithm and the best classification technique to achieve superior detection rates with low training time, accurately classifying attacks on a given dataset. It enables agents to rapidly learn how to select the correct policy for each action. The efficacy of the proposed approach is evaluated through experiments on three datasets.

## 3. Existing works

To enhance the performance of IoT networks in intrusion detection, researchers have required to extract attack features across multiple aspects to classify attacks [33]. In current research, there is a growing focus on applying chaos theory to evolutionary algorithms, which describe the chaotic behaviour of nonlinear dynamic systems that are highly dependent on initial conditions. The practical application of chaos in evolutionary algorithms has increased significantly, particularly in cyber detection, where researchers have employed nature-inspired techniques to enhance detection rates. Researchers have established relationships between suitable intelligent techniques and each intrusion class to improve performance. However, the literature offers divergent findings on the effectiveness of different approaches for detecting different classes of intrusions.

Researchers have recently been interested in leveraging different algorithms and strategies through hybrid mechanisms [34]. Differential evolution (DE) is one of the most effective evolutionary algorithms for balancing exploration and exploitation, and for perceiving related features that often exhibit aggregation. Typically, mutation and crossover operators are utilised to enhance exploitation, with the mutation operator adding diverse information into the population and improving exploration. Various GOA-based techniques have been proposed in the literature to defend against modern attacks, but they have limitations. Thus, a multi-population searching strategy is utilised to improve population diversity and speed up convergence.

### 3.1. Grasshopper optimization algorithm

To address complex real-world optimization problems, researchers have proposed numerous wrapper-based feature selection methods integrated with various evolutionary algorithms. Recently, algorithms based on swarm behaviour have become a significant movement in modern metaheuristic optimization. These algorithms strategically balance exploration and exploitation, iteratively evolving a population of initially random solutions toward a global optimum [35]. Among these current metaheuristics, Grasshopper Optimisation Algorithm (GOA) has developed as one of the most powerful metaheuristics algorithms in recent years. It is inspired by the natural food-searching behaviours and complex social interactions observed within swarms of grasshoppers in nature [36]. GOA method entails creating potential solutions to generate the principal artificial grasshopper. After evaluating each candidate grasshopper's fitness values, the top search grasshopper is chosen as leader or target. Every grasshopper starts toward the target grasshopper as it draws in neighbouring individuals. Based on exploration and exploitation, this algorithm draws inspiration from nature and separates the search process into two tendencies [37]. During exploration, search agents move quickly, while during exploitation, they typically move locally. Grasshoppers are inherently adept at both of these tasks. A grasshopper swarm behaviour mathematical model was presented in [38], where the author designed using Eq. (1) to solve various optimisation problems.

$$Y_i^d = cz \left\{ \sum_{\substack{j=1 \\ j \neq i}}^{n} z(\frac{ul_d - ll_d}{2}) c.sf(|Y_j^d - Y_i^d|)(\frac{Y_j - Y_i}{d_{ij}}) \right\} + \hat{T}_d$$

(1)

Where $sf$ is a function to define the strength of social forces, c is a decreasing coefficient that balances exploration and exploitation, the number of grasshoppers, and distance between $j^{th}$ and $i^{th}$ Grasshoppers are denoted by n and $d_{ij}$ respectively and $\hat{d}_{ij} = \frac{Y_j - Y_i}{d_{ij}}$ is unit vector from $i^{th}$ grasshopper to $j^{th}$ grasshopper as shown in Eq. (1). Here, $ul_d$ and $ll_d$ are the upper bound and lower bound in $D^{th}$ dimension, $j^{th}$ and $i^{th}$ positions of the grasshopper and value of target are represented by $Y_j^d$, $Y_i^d$ and $\hat{T}_d$ in $D^{th}$ dimension respectively whereas $cz$ denotes a decreasing constant to reduce the comfort zone, repulsion zone, and attraction zone. The algorithm's most critical control parameter, cz, is updated using Eq. (2).

$$cz = cz_{max} - iter\left(\frac{cz_{max} - cz_{min}}{t_{max}}\right)$$

(2)

Where $cz_{max}$ and $cz_{min}$ the maximum and minimum values, respectively, designate the current iteration $iter$, and $t_{max}$ is the maximum iteration number. The grasshopper's movement around the target is decreased by the first cz in Eq. (2). The second cz is thought to stop grasshoppers from moving too much. The GOA scheme flowchart is shown in Fig 1.

### 3.2. Support vector machine

Support Vector Machines (SVMs) are highly effective algorithms for classification problems, and their ability to handle high-dimensional datasets makes them applicable to diverse domains such as image and intrusion detection. SVMs demonstrate robustness in scenarios where the number of features surpasses the number of samples, enabling effective extraction of intricate patterns from complex data [39,40]. SVMs surpass in scenarios involving intricate decision boundaries and datasets with high-dimensional features by maximising the marginal distance between the hyperplane and the

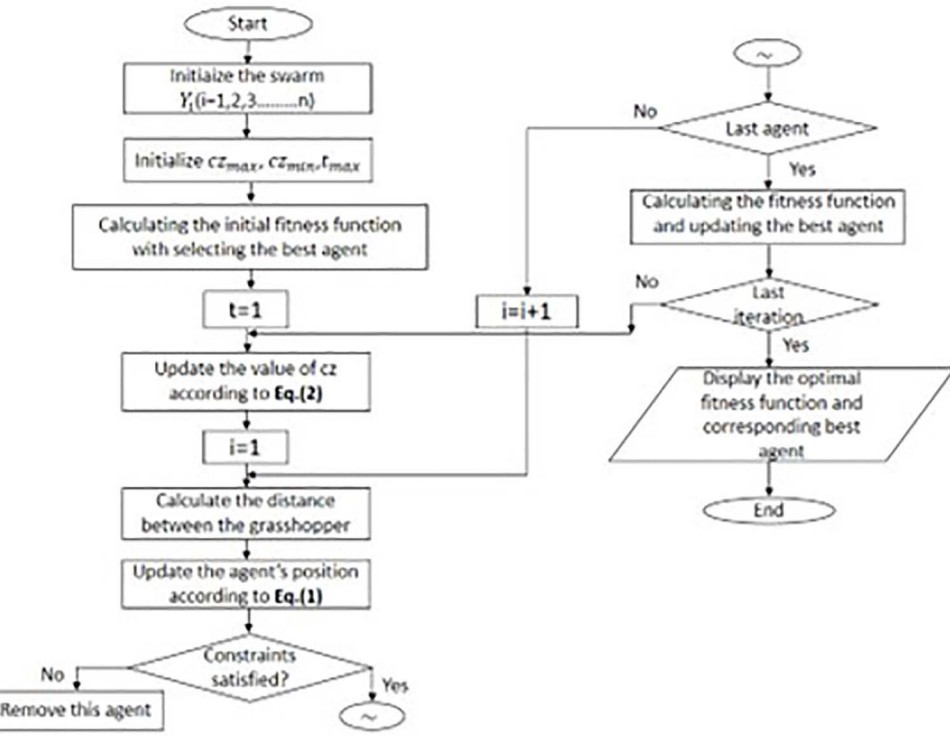

**Fig 1. Flow chart of the basic Grasshopper optimisation algorithm.**

corresponding data instances [41]. By focusing on support vectors, which are data points closest to the hyperplane, they can efficiently categorise new samples based on their relative position to the decision boundary. Notably, SVMs exhibit resilience to over-fitting by striking a balance between fitting the training data and generalising to novel samples. The hyperplane that can separate the classes is obtained as enumerated in Eq. (3).

$$w^t.x + b = 0 \tag{3}$$

Where $w$ and $b$ are the coefficient vector and the bias, respectively, most of the records are separated linearly in some applied cases, but some irregular instances that cannot be separated may exist. To overcome this problem, SVM can achieve a maximum margin by solving the following optimisation problem, as depicted in Eq. (4).

$$\left( \frac{\|w^2\|}{2} + C \sum_{i=1}^{l} \xi_i \right) \tag{4}$$

$$s.t. \left\{ y_i \left( w.x_i + b \right) \geq 1 - \xi_i \; \xi > 0 \right\}$$

To simplify the calculation, the above equation can be transformed into its corresponding dual problem using Lagrangian multipliers $\beta_i > 0$ ($i = 1, 2, \ldots, l$), as in Eq. (5).

$$max_\beta \left[ \sum_{i=1}^{l} \beta_i - \frac{1}{2} \sum_{i=1}^{l} \sum_{j=1}^{l} \beta_i \beta_j y_i y_j (x_i x_j) \right] \tag{5}$$

$$s.t., \sum_{i=1}^{l} \beta_i y_i = 0, \; \beta_i \geq 0$$

This problem can be solved by using quadratic programming. The optimal value $\beta$ is denoted by $\beta^* = [\beta_1^*, \beta_2^*, \ldots, \beta_l^*]^T$ and the best values of $w^*$ and $b^*$ An can be obtained as follows in Eqs. (6–7).

$$w^* = \sum_{i=1}^{l} \beta_i^* x_i y_i \tag{6}$$

$$b^* = -\frac{1}{2} w^* (x_r + x_s) \tag{7}$$

Where $x_r$ and $x_s$ are any pair of support vectors in the two classes. The final linear discriminant function can be achieved by Eq. (8).

$$f(x) = sgn \left( \sum_{i=1}^{l} \beta_i^* y_i (x.x_i) + b^* \right) \tag{8}$$

To adjust linear methods for non-linear cases, SVMs employ a kernel function that maps data points into a higher-dimensional space, facilitating easier and more accurate classification. The classification function can be transformed as follows in Eq. (9) to perform the mapping.

$$f(x) = sgn \left( \sum_{i=1}^{l} \beta_i^* y_i \phi(x_i)^t \phi(x) + b^* \right)$$

(9)

Where the form of $\phi(x_i)^t \phi(x)$ the feature space is represented $x_i^t x$ in the input space, and the map of forming $\phi(x_i)$a Be obtained implicitly according to the selected kernel. $K(x_i, x_j) = \phi(x_i) \phi(x_j)$ .The decision function can be constructed as follows in Eq. (10).

$$f(x) = sgn \left( \sum_{i=1}^{l} \beta_i^* y_i k(x_i, x) + b^* \right)$$

(10)

The Gaussian kernel, also known as the Radial-Basis Function (RBF), is the most widely adopted kernel in SVM. In this study, we applied the RBF to SVM, and the RBF kernel is formulated as follows in Eq. (11).

$$k(x_i, x) = exp(-\gamma \|x - x_i\|^2)$$

(11)

It $\gamma$ is defined ahead of time and is a parameter controlling kernel width.

### 3.3. Chaotic multi-population GOA with DE: CMGODE

This paper proposes a novel Chaotic Multi-population Grasshopper Optimization Algorithm integrated with Differential Evolution (CMGODE) to address critical limitations in securing IoT networks. While machine learning and evolutionary algorithm-based approaches are essential for classifying network traffic as normal or malicious, and swarm intelligence algorithms outshine at detecting novel attacks, conventional methods like basic GOA are often hindered by slow convergence and tendency to settle at local optima. To overcome these shortcomings, proposed CMGODE model enhances the fundamental GOA framework through three key innovations: the incorporation of chaos theory to improve initial population diversity and prevent premature convergence, a multi-population strategy to better manage high-dimensional IDS data and balance exploration with exploitation, and the hybridization with differential evolution to refine solution quality and accelerate convergence. The primary objective of this hybrid model is to achieve efficient and accurate network traffic classification by identifying an optimal minimal subset of features, thereby maximizing detection accuracy while minimizing computational time and resource consumption for real-world IoT security applications.

**3.3.1. Chaos map.** The literature on chaos theory has viewed several developments in the past few decades [34]. Chaos, characterised by its sensitivity to initial conditions, has found extensive applications in various domains involving non-linear dynamic systems. Due to its randomness and regularity, chaos has been successfully incorporated with several EAs to achieve global optimal solutions in various engineering optimisation problems. In heuristic algorithms, the initial population is generated randomly, and enhancing its diversity can help prevent entrapment in local optima and premature convergence. In recent years, researchers have utilised chaos theory to optimise and improve different evolutionary and swarm intelligence techniques, as discussed in the literature review section.

The basis of this approach is the hypothesis that an arbitrary distribution can be deemed as uniform. Nevertheless, the convergence speed and accuracy of these algorithms often rely to some degree on the quality of the initial population [42]. It is widely recognised that a good initial population can lead to a rapid convergence to the global optimum. To enhance the performance of the proposed algorithm, this study introduces the use of a chaotic sequence to generate the initial population. In this regard, the chaotic sequence employed by CMGODE is generated through the logistic map and is defined in Eq. (12).

$$X_{i+1} = \mu X_i \times (1 - X_i) \quad i = 0, 1, 2, \ldots n$$

(12)

Where $\mu$ and $X_i$ show the control parameter and $i^{th}$ chaotic variable, respectively, to generate the chaotic sequence, the value of $\mu$ is set to 4, which is also called the bifurcation coefficient; additionally, let $0 < X_0 < 1$, and $X_0 \neq 0.25, 0.5, 0.75, 1$.

### 3.3.2. Multi-population scheme.

Inspired by natural evolution processes, researchers have employed multi-population strategies to enhance EA search capability and overcome specific challenges in Intrusion Detection Systems (IDS) [7]. Firstly, this method decomposes the initial population into several small sub-populations. Then, these sub-populations are evaluated in parallel using GOA. Finally, to find optimal solutions, different sub-populations collaborate to search other local areas. The size of the population can significantly influence the average calculation time of EAs, transforming exponential time into polynomial time under certain circumstances. Studies have shown that introducing a population can increase the probability of first strike [43]. Therefore, adjusting the population size to an appropriate value is advisable to minimise calculation time and improve the striking probability. A practical approach to managing population size, which reduces calculation time without compromising diversity, involves dividing a large initial population into multiple smaller sub-populations and performing parallel executions.

Previous literature has outlines two principal approaches in this area of fundamental research. The first approach involves a multi-population scheme, where each subpopulation follows a distinct strategy. However, one specific sub-population maintains the same search strategy throughout the process. The second approach is a single-population scheme that incorporates multiple strategies and parameters. A set of parameters is selected as the current parameters based on specific rules. This approach often switches between multiple evolutionary strategies, with each evolution using only one strategy. It is possible to adjust the algorithm parameters to leverage parallel computing and reduce computational time, ensuring compatibility with parallel processing. Keeping this in mind, the algorithm proposed in this study divides the entire population into equal $N$ sub-populations based on fitness values. Each subpopulation size is set to $Pop/N$, and mutation strategies and control parameters are applied. Notably, Pop's overall population size remains, thereby preserving population diversity and preventing premature convergence.

### 3.3.3. Information communication scheme.

In this study, we have utilised cross-sub-population migration as an information communication scheme that replaces solutions based on the solution quality implemented by probabilistic choosing. Then, we identify pairs of sub-populations suitable for migration based on their respective levels. We use the distance levels to decide which sub-populations to migrate to or from [44]. The rationale behind this approach is that sub-populations exhibiting higher fitness distance are more likely to benefit from migration than those with lower distance. Once the pair of subpopulations to migrate from is determined, we can calculate the distance between the solutions being replaced, and then the subpopulation will be selected. The utilisation of distance is rooted in diversity, recognising that a more diverse population offers increased opportunities for discovering optimal solutions.

## 3.4. Differential evolution strategy

The DE strategy is employed to enhance both the local search capability of the fundamental GOA and the quality of the solution produced. A multi-population scheme is initially utilised to implement the DE strategy, where a population is processed to generate a new population through various operations, including mutation, crossover, and selection. The specific details of each operation are elaborated upon below.

### 3.4.1. Mutation operation.

The DE strategy introduces individual variation by utilising a basic DE strategy. The process begins by randomly selecting three distinct swarms from the entire population. Next, the difference between the two of these individuals is scaled. Finally, the scaled difference vector is combined with the third swarm to produce the final variant individually. The mutation operation function is defined in Eq. (13).

$$V^i(K) = X^{rn_1}(K) + F.(X^{rn_2}(K) - X^{rn_3}(K)) \qquad (13)$$

where $i \neq rn_1 \neq rn_2 \neq rn_3$, $i = 1, 2, \ldots, n$, K represent the iteration number and F denotes alling factor $\in [0, 2]$. In this paper, the scaling factor was randomly selected. For each individual, r is performed in which a value between 0.1 and 0.9 is randomly chosen and used as the scaling factor's value.

**3.4.2. Crossover operation.** The crossover operation within DE plays a vital role in generating novel candidate solutions by merging information from different individuals. The specific functions are described in Eq. (14):

$$U_j^i (K) = \{V_j^i(K), \qquad if \ rand(0, 1) \leq Cr\|j$$
$$= j_{rand} X_j^i (K), \qquad otherwise \tag{14}$$

where $j = 1, 2, \ldots, D$, Cr is the crossover probability $\in [0, 1]$ and $j_{rand}$ signifies a randomly selected index between 1 and D. At least one dimension of this design causes the value of $Vj^i (K)$ Cr to come from the again generated swarm $U_j^i (K)$.

**3.4.3. Selection operation.** The selection operation is performed in the DE strategy, and the swarm $Ui^i (K)$ generated by the crossover operation is compared with the original swarm $X^i (K)$ to determine the swarm $X^i (K+1)$ of the next generation. The role of the selection operation is defined in Eq. (15).

$$X^i (K+1) = \{U^i (K), f\left(Ui^i (K)\right) \leq f\left(X^i (K)\right) X^i (K), \ k = 1, 2, \ldots, N. \tag{15}$$

## 3.5. Fitness function

The fitness function refers to a process used to assess the quality of solutions. In this context, the fitness function evaluates a solution, which represents a subset of selected features, based on its True Positive Rate (TPR), False Positive Rate (FPR), and the number of features it encompasses. Including the number of features in the fitness function allows us to prioritise excluding any features that do not significantly impact the TPR or FPR. Eq. (16) presents the formula employed to evaluate the fitness of the grasshopper or solution.

$$F = \frac{S_f}{N_f} + \frac{FP}{FP + TN} + \frac{TP}{TP + FN} \tag{16}$$

Where $S_f$ denotes the number of selected features, and $N_f$ is the total number of features.

## 3.6. Hypotheses about proposed algorithm

Although, efficacy of basic grasshopper optimization algorithm is recognized for its relatively fast convergence and simple structure when applied to complex, high-dimensional problems, a key limitation arises as its frequent inability to effectively balance the exploration and exploitation phases. This imbalance often results in premature convergence to local optima. To address this fundamental challenge, this paper expanding the basic GOA by integrating chaotic multi-population strategy with differential evolution mechanism. This hybridized Chaotic Multi-population GOA with DE (CMGODE) is specifically designed to enhance performance in the critical task of significant feature selection for IoT security. The introduced multi-population topology is pivotal, as it increases swarm diversity and refines the balance between exploitation and exploration by strategically controlling the rate and direction of information exchange among co-evolving subpopulations. The ultimate aim of this integration is to develop a more robust optimizer that improves the accurate detection of malicious attacks within IoT network environments.

The proposed approach relies on multiple swarms working together through cross-sub-population migration as an information communication scheme. Initially, the population is divided into numerous sub-swarms during the early stage, and as the evolutionary process unfolds, the number of sub-swarms is periodically reduced. This strategy helps maintain

a balance between early-stage exploration and late-stage exploitation. By leveraging historical information from the search process, proposed approach can detect potential swarms where the population becomes trapped in local optima. This enables the population to discontinuity free from the current local optima, enhancing its exploration capability. Our CMGODE divides the entire population into subpopulations. This ground-breaking mechanism leverages chaos theory to promote efficient exploitation of potential local regions while effectively improving the quality of neighbourhood structures and preserving population diversity. Additionally, the migration mechanism facilitates the sharing of valuable information among subpopulations throughout the search process. Furthermore, integrating the differential evolution strategy into the GOA enhances the local search capabilities of the proposed variant. This integration boosts the fundamental GOA algorithm ability to explore local optima and significantly improves the overall quality of the generated solutions. The main aim is to strike a balance between exploitation and exploration. The proposed techniques and the detailed pseudo code for CMGODE are elaborated upon in the subsequent sections.

## Algorithm: CMGODE

```
Initialization population size 'P,' number of sub_population' k', size of sub_ population' S_sub = P/k
,Max number of iteration 't'_max, target ' T̂_d and termination criterion;
Generate the initial population of GOA using the chaos concept and set current generation i = 1
Each dimension of the population space has divided the population into k equal size
sub_population P_1, P_2, P_3..........P_k
While (i <t_max)
i  =  i  +1;
For j= 1→k
  Estimate the fitness value of each sub_population
For l= 1→ round (P/k)
  P'_j,l →Update the position of P_j,l candidate solution according to Eq. (1)
End For
If (P'_j,l) better than O(P_j,l)
  Update the target ^T_d
End If
End For
Merge the entire sub_population (P_1, P_2, P_3..........P_k)to P
If F_best(i+1) better than F_best(i)
 k = k+1;
Else If k>1
 k = k−1;
End If
Apply DE technique
End While
```

Our method is illustrated in the flowchart shown in Fig 2. After setting the initial parameter values, a random population of candidates is produced and assessed. The population is then split into $m$ smaller groups, each using GOA algorithm. If the problem changes, the algorithm adjusts the sub-population dimension and checks the halting condition by computing the change in terms of fitness value. The method terminates and provides the optimal solution if the designated termination condition is met. If not, it re-divides the population into $m$ sub-populations, applies the selection process, creates a new iteration, updates the population, and merges all sub-populations. Below is more information on the primary steps.

- The population size (*Pop*), number of sub-swarms (k), sub-swarm size (S_sub), the termination criterion (F_max), and the maximum number of iterations (t_max) are the initial values of the main parameters of our approach. Its definition defines the maximum number of function evaluations. Assume that the fitness function is F, and create the starting population, Pop.

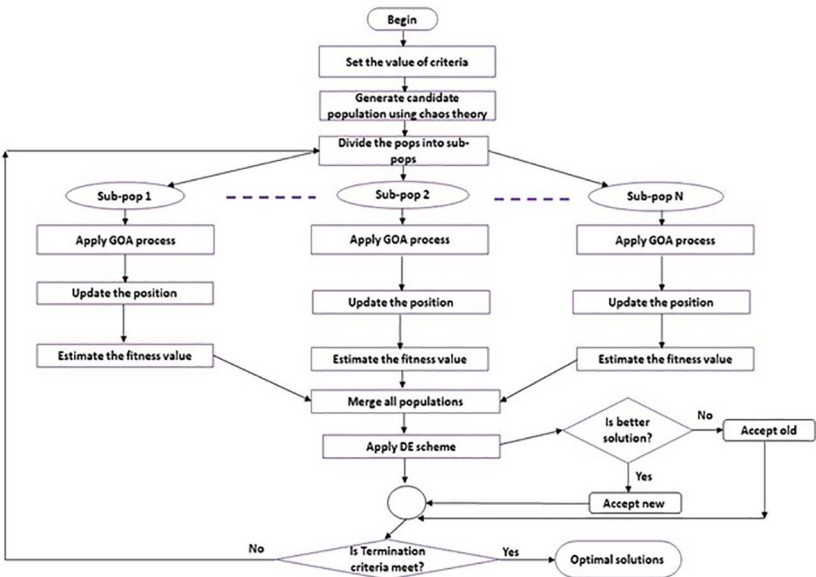

**Fig 2. Flow chart of the proposed framework.**

- The population *of Pop* is divided into equal groups. Determine the fitness value of each subpopulation independently. Each subpopulation independently uses the GOA algorithm to modify the solutions within its groupings. The altered solutions may be approved if they outperform the original ones. After this, the entire subpopulation is combined.

- The quality of the current best solution for the entire population (F(migrate)) and the previous best solution for the whole population (F(old)) are then compared. The search process exploration feature is improved if the value of F(migrate) exceeds that of F(old). On the other hand, the algorithm becomes less exploratory and more exploitative if F(migrate) is not superior to F(old). The quality of the resulting solution is improved by implementing a migration plan.

- After every iteration, the DE algorithm updates the best individual. In this instance, the multi-population GOA algorithm exploitation search skills are intended to be improved by the DE algorithm. Verify the termination condition.

- Close the loop once the maximum number of assessments has been completed. If the maximum number of evaluations has been reached, end the loop and report the best value. Otherwise, follow the following steps:

- Again, the population's fitness will be recalculated, and we will start from the step above.

- Select the best-generated solutions from the entire population.

   In response to the research above, we developed an algorithm, CMGODE that handles anomalies described in the proposed approach and incorporates an effective wrapper technique. The best features are then chosen using GOA and sent into the SVM algorithm to build the model. To suggest the best algorithm for identifying network intrusion in the context of an Internet of Things network, we examined the performance of the SVM algorithm. The algorithm efficacy is assessed using accuracy, detection rate, false alarm rate, precision, and F-measure compared to current state-of-the-art methods.

   **3.6.1. Computational complexity analysis.** A foundational concept in computational study is algorithmic complexity, which quantifies the resources primarily time and space required by an algorithm to solve a problem as a function of its input size. The computational complexity of our chaotic multi-population grasshopper optimization algorithm with

differential evolution (CMGODE) is influenced by several key factors: population size, dimensionality of the feature space, overhead from chaos-inspired initialization and updates, multi-population strategy managing sub-populations, and integrated Differential Evolution operations. Understanding CMGODE complexity necessitates a breakdown of its major computational steps initialization, fitness evaluation, position updates with chaotic maps, inter-subpopulation communication, and DE-based refinement to analyze their individual and combined impact on the algorithm overall time and space efficiency.

Suppose $pop$ is the population size, $D$ is the problem dimension, and $tm$ denotes the maximum number of iterations. The standard Grasshopper Optimization Algorithm (GOA) comprises three main phases: population initialization, fitness evaluation, and iterative position updating. Initializing pop grasshoppers in a D-dimensional search space has a complexity of $O(pop \cdot D)$. Evaluating fitness depends on the cost of the objective function, denoted as $£$, giving a complexity of $O(pop \cdot £)$. Each iteration updates grasshopper positions using attraction-repulsion forces, requiring $O(pop \cdot D)$ operations. Since the algorithm runs for tm iterations, the overall complexity of standard GOA is $O(tm \cdot pop \cdot D)$.

Chaotic initialization in CMGODE improves solution diversity without increasing asymptotic complexity; it remains $O(pop \cdot D)$ despite the added chaotic mapping. The multi-population strategy divides the population into $subP$ subpopulations that evolve independently, incurring a division cost of $O(pop)$ and a per-iteration cost of $O(tm \cdot (pop/subP) \cdot D)$. Migration between subpopulations adds a minor $O(pop)$ overhead. Integration of Differential Evolution (DE) introduces mutation, crossover, and selection steps, each contributing $O(pop \cdot D)$ per iteration. Consequently, the additional operations from chaos, multi-population, and DE do not raise the overall order of complexity. Thus, computational steps introduced by chaos operations, multi-population strategies, and DE refinements, the final computational complexity of CMGODE remains, which can be expressed as $O[(tm*pop*D)+O(tm*pop)+(pop*D+£)]$.

# 4. Experimental results and discussion

This section presents the experimental results of the proposed approach compared to existing state-of-the-art methods, evaluated on two real-world benchmark datasets: BoT-IoT and UNSW-NB15. The proposed approach employs chaotic multi-population GOA with DE scheme to select relevant features that contribute to identifying malicious activity in IoT network environments. The performance evaluation of the CMGODE method utilizes SVM as a fitness function for attack prediction. To assess their effectiveness, a comparison is made against GA, DE, ABC, PSO, and GOA in terms of Recall, F-measure, False Positive Rate (FPR), Precision, Accuracy, and Computational Time. The results demonstrate that the CMGODE approach, combined with the SVM classifier, outperforms other state-of-the-art techniques.

## 4.1. Data pre-processing

For feature selection and model training, pre-processing is necessary to clean and format the raw data so that it is available in an appropriate format before ML models are trained. The initial dataset may be pre-processed using data transformation and cleansing methods to build an intrusion detection classification model. The proposed model initial pre-processing phase improves robustness by removing variations and noise. This can be achieved through data cleaning and normalisation techniques. Unclean data is defined as missing, distorted, or incomplete information, resulting in poor performance in data processing and analysis due to the original data being fragmented and lacking internal rules. As a result, unclean data can be cleaned and converted into data that meets data quality requirements.

### 4.1.1. Data collection.
Data collection is the counterpart and a necessary stage for an intrusion detection system. The design and efficacy of an IDS are determined by two factors: type of data source and the position from which the data is collected. To provide the most appropriate security for the host or target network, this study proposes to test our approach on two IDS-based web network. Throughout the training phase, data samples are classified using web-level procedures and labelled based on domain knowledge. However, the data collected during the testing phase are merely classified according to the protocol kinds.

**4.1.2. Data transferring.** The main challenges of data cleansing are distortion, misfits, missing values, and inaccurate data. The same constant can replace or eliminate the present special symbols and garbled codes, and fill in the missing data. Categorical features were encoded using an encoding scheme. In encoding, each category is assigned a numerical label. In this section, each record in the input datasets is denoted as a matrix of real values for the trained classifier. Accordingly, every representative symbol in a dataset initially becomes a numerical value.

**4.1.3. Data normalization.** During this section, we collect and transform the data during the data collection and data transformation phases. It is utilised for our experiments to evaluate the necessary features in IDS data, namely BoT-IoT and UNSW-NB15. This process ensures that the model can process the categorical data. After encoding, we normalise BoT-IoT and UNSW-NB15 to [0,1] to increase the model's resilience and remove the impact of various features taking on a broad range of values. Eq. (17) is the expression for the Normalisation min-max method:

$$x_n = \frac{x - x_{min}}{x_{max} - x_{min}}$$

(17)

Each feature in Eq. (17) has a minimum value denoted by $x_{min}$, a value denoted by x, and a maximum value denoted by $x_{max}$. To mitigate the impact of class imbalance, additional synthetic samples are generated for the minority classes based on their distributions using synthetic minority over-sampling technique, thereby balancing the class distribution before model building [25]. After finishing the pre-processing stages, both datasets are prepared for machine learning and EA and ML techniques.

**4.1.4. Feature selection process.** Once a substantial number of subsets are chosen, the optimal sets are obtained during the classifier training step and used by SVM. Select the best features from all of the attributes in the BoT-IoT and UNSW-NB15 datasets. Because SVM can easily deal with classification challenges and has multiple classes, we utilise an SVM classifier to differentiate the instances from the datasets. We have determined that the parameters with the highest detection accuracy for classification are the best fit. Our chaos multi-population GOA, combined with the differential evolution optimisation algorithm, determines the optimal intrusion detection function using an SVM classifier to identify the best feature sets. Some existing optimisation algorithms employed for feature selection have drawbacks, such as slow convergence time, premature convergence, and sensitivity issues. CMGODE seeks to strike a balance between exploration and exploitation. This balance enables the algorithm to explore the solution space more efficiently and avoid premature convergence to suboptimal solutions. Hence, it is utilised to pick optimal features. Furthermore, this approach optimises the parameters and minimises the error, demonstrating its independence in various processes.

## 4.2. Description of datasets

**4.2.1. UNSW-NB15 dataset.** The UNSW-NB15 dataset is a well-known benchmark dataset extensively used in network security and intrusion detection [16]. It was created to mimic real-world network environments, comprising diverse network traffic data that encompasses both normal and malicious activities. With a focus on network traffic, the UNSW-NB15 dataset encompasses various features that capture crucial aspects, including source and destination IP addresses, port numbers, protocol types, and payload characteristics. This dataset contains approximately 25,40,044 data samples, each consisting of 49 features. The training dataset comprises 175,341 records, while the testing dataset contains 82,332 records. The apportioned informational collection has just 43 features, with the class label expelling six features from the entire dataset.

**4.2.2. BoT-IoT dataset.** The BoT-IoT dataset is widely recognised and extensively used as a benchmark dataset in IoT security. It was collected from the Cyber Range Lab of the Centre of UNSW Canberra Cyber [29]. It has been carefully curated to replicate real-world IoT network environments, offering comprehensive network traffic data collection. This dataset encompasses a diverse range of features, including source and destination IP addresses, port numbers, protocol

types, and payload characteristics. The explanations of features can be seen in [45]. This paper uses 10% of the collected dataset to train the model. In the datasets, attack samples were labelled with a value of 1, and standard samples were labelled with a value of 0 to train and validate machine learning models through binary classification.

## 4.3. Experimental setup

To evaluate the effectiveness of our approach on IoT datasets, all experiments in this study are conducted using MAT-LAB R2018 on Windows 10, with an Intel Core i7-2.4 GHz processor and 16 GB of RAM. Normalisation is a scaling-down transformation that projects a feature onto a normalised range [46]. We use it for our experiments. Standard parameter settings were used in this paper to guarantee a fair comparison. Consequently, all optimisation techniques, including GA, DE, ABC, PSO, and GOA, had the same population size and number of iterations. Notably, when the maximum number of iterations was set to 100 and the population size was set to 90, the behaviour of these algorithms, especially GOA and the suggested method, exhibited ideal performance.

## 4.4. Performance measures

Different criteria can be used to evaluate the wrapper methods, depending on the specific application. When evaluating intrusion detection systems, researchers often use performance indicators such as detection rate (DR) and false positive rate (FPR). In this section, we establish a set of performance indicators to evaluate the effectiveness of our technique. Some of these measurements include AUC (Area under the curve), F-measure, accuracy, recall, precision, and false positive rate. The performance metrics are displayed as below:

$$Accuracy = \frac{TN + TP}{TP + TN + FP + FN} \quad Detection\ rate(DR) = \ Recall\ (Re) = \frac{TP}{TP + FN} \quad Precision\ (Pr) \ = \frac{TP}{TP + FP}$$

$$F_{measure} = \frac{2 * Pr * Re}{Pr + Re} \quad AUC = \frac{(Sensitivity + \ Specificity)}{2} \quad False\ positive\ rate\ (FPR) = \frac{FP}{FP + TN}$$

False Positive, False Negative, True Positive, and True Negative, FP, FN, TP, and TN, are in the IDS data.

## 4.5. Results and discussion

Our method is assessed in this section using the UNSW-NB15 and BoT-IoT datasets. This study compares state-of-the-art approaches for network intrusion detection systems used in IoT scenarios, such as GA, DE, ABC, PSO, and GOA, with the chaotic multi-population GOA with DE scheme implemented using an SVM engine. The assessment considered accuracy, F-score, AUC, FPR, and TPR. The suggested method outperforms other EAs selected from cutting-edge, pertinent research on the BoT-IoT and UNSW-NB15 datasets, utilising various modelling tools, according to the previously cited literature [47]. The results shown are all averages over ten runs.

### 4.5.1. Comparison of feature selection methods using SVM.
When a complete evaluation is infeasible, Evolutionary Algorithms (EAs) such as Genetic Algorithm (GA), Particle Swarm Optimization (PSO), Differential Evolution (DE), Artificial Bee Colony (ABC), and Grasshopper Optimization Algorithm (GOA) offer a key advantage. They facilitate the selection of an optimal feature subset, which enhances overall model performance. The proposed method leverages these algorithms to identify the most relevant features from two Internet of Things datasets. GOA and DE methods aid with optimisation problems in binary attack and multi-class classification by utilising SVM learning variations in conjunction with other evolutionary strategies.

The outcomes of running our system on the UNSW-NB15 and BoT-IoT datasets are shown in this section. Consequently, Table 1 presents the outcomes of the proposed system multi-classification for the BoT-IoT and UNSW-NB15

**Table 1. Average Classification performance using wrapper methods on BoT-IoT and UNSW-NB15.**

| BOT-IOT | | | | | | | UNSW-NB15 | | | | | |
|---|---|---|---|---|---|---|---|---|---|---|---|---|
| Classifier | Measure | GA | PSO | DE | ABC | GOA | Proposed | GA | PSO | DE | ABC | GOA | Proposed |
| SVM-R | Accuracy | 85.34 | 85.93 | 87.67 | 85.62 | 91.34 | 99.79 | 98.23 | 98.55 | 98.51 | 98.02 | 98.01 | 99.53 |
| | DR | 84.91 | 85.72 | 87.91 | 85.43 | 90.67 | 98.92 | 97.78 | 98.09 | 98.27 | 98.71 | 98.12 | 99.12 |
| | Precision | 83.37 | 84.99 | 86.54 | 85.21 | 89.23 | 99.09 | 97.34 | 97.52 | 98.12 | 97.03 | 96.99 | 97.61 |
| | F-measure | 84.01 | 85.05 | 86.99 | 84.99 | 90.37 | 98.89 | 98.23 | 98.72 | 98.63 | 96.82 | 97.92 | 97.59 |
| SVM-P | Accuracy | 84.21 | 83.98 | 84.42 | 83.99 | 88.76 | 98.23 | 96.76 | 96.82 | 96.12 | 96.71 | 96.82 | 96.03 |
| | DR | 84.11 | 83.99 | 85.71 | 83.62 | 86.72 | 98.11 | 96.67 | 96.23 | 96.73 | 96.82 | 97.01 | 95.04 |
| | Precision | 84.02 | 83.72 | 84.79 | 84.03 | 89.33 | 97.54 | 96.52 | 94.32 | 95.72 | 97.28 | 97.00 | 97.04 |
| | F-measure | 81.34 | 84.90 | 86.53 | 85.11 | 86.83 | 97.62 | 95.61 | 96.72 | 96.82 | 95.88 | 95.82 | 96.78 |
| SVM-L | Accuracy | 82.98 | 81.43 | 83.43 | 83.54 | 84.87 | 95.81 | 92.72 | 91.34 | 90.22 | 91.81 | 92.72 | 93.34 |
| | DR | 82.11 | 80.52 | 82.42 | 82.79 | 83.02 | 94.52 | 93.82 | 93.12 | 94.01 | 93.89 | 93.67 | 93.74 |
| | Precision | 81.52 | 80.62 | 81.43 | 81.65 | 82.98 | 95.15 | 92.63 | 92.01 | 92.60 | 92.76 | 92.01 | 93.89 |
| | F-measure | 80.23 | 79.04 | 80.43 | 80.53 | 81.66 | 93.54 | 91.23 | 92.73 | 92.82 | 92.90 | 92.67 | 94.89 |

datasets, accounting for various SVM classifier configurations and EA-based feature selection methods. The findings of evaluating the suggested system by contrasting it with five conventional EAs-based studies are shown in this section. These findings are presented using the average metrics from ten stratified ten-fold cross-validation runs.

Table 1 results show that the SVM-R classifier outperformed the other SVM classifiers. Moreover, the proposed SVM-R outperformed the different methods, yielding superior accuracy and F-measure outcomes in both datasets when accuracy, detection rate, F-measure, and precision (as percentages) were considered.

Table 2 compares the proposed method with the current SVM-R strategy in terms of training, testing accuracy, and estimated run time. These conclusions were drawn based on ten trials using the BoT-IoT and UNSW-NB15 datasets. According to the data, the proposed method consistently yields more reliable and stable results than its five rivals.

**4.5.2. Comparative studies in terms of performance.** This section compares the results and comments of the proposed method with those of five different wrapper strategies that utilise the SVM-R classifier. Table 2 displays the training accuracy rate (98.52) and testing accuracy rate (98.43) for the proposed system for multi-classification of the BoT-IoT dataset, as well as the execution time 373.5 (secs). Additional investigation has been conducted to categorise the UNSW-NB15 dataset, as indicated in Table 2. It uses tenfold cross-validation to summarise the performance study, concentrating on both datasets. The results of the suggested method and other approaches are presented in the table, considering factors such as accuracy, detection rate, false positive rate, training execution time, and testing execution time. Figs 3 and 4, illustrates that our approach can outperform other cutting-edge EAs in terms of performance.

The results indicate that proposed technique yields better operational outcomes in classification performance when compared to earlier detection procedures. As the table illustrates, it achieves both an optimal detection rate and a low false positive rate on the UNSW-NB15 dataset, as well as an optimal detection rate and a low false positive rate on the BoT-IoT dataset. In addition, compared to other approaches, the proposed solution significantly reduces training and testing timeframes.

Simulations are conducted to evaluate the performance of the proposed method using different wrapper strategies for identifying various types of attacks over time. Using the BoT-IoT and UNSW-NB15 datasets, Figs 5-8 illustrate the multiclass classification performance of the proposed intrusion detection technique, providing more insight into the advantages of the suggested approach. The proposed method achieves a high identification rate on the BoT-IoT and UNSW-NB15

**Table 2. Evaluation of BoT-IoT and UNSW-NB15 with different scales.**

| Data | Measures | GA | PSO | DE | ABC | GOA | Proposed |
|------|----------|-----|------|-----|-----|-----|----------|
| BoT-IoT | Train Accuracy | 96.82±1.19 | 93.90±1.21 | 94.52±1.35 | 95.23±1.37 | 96.00±1.20 | 98.52±0.97 |
| | Test Accuracy | 94.53±1.03 | 92.78±1.05 | 93.02±1.04 | 94.21±1.16 | 94.78±1.07 | 98.43±0.81 |
| | Runtime | 734.73 | 812.72 | 888.23 | 912.02 | 452.87 | 373.5 |
| UNSW-NB15 | Train Accuracy | 97.02±1.29 | 96.89±1.31 | 96.72±1.24 | 96.03±1.38 | 96.82±1.29 | 98.98±1.15 |
| | Test Accuracy | 96.54±1.01 | 95.83±1.29 | 95.73±1.31 | 95.78±1.17 | 95.73±1.36 | 98.11±0.94 |
| | Runtime | 893.81 | 745.02 | 678.34 | 589.63 | 661.93 | 444.73 |

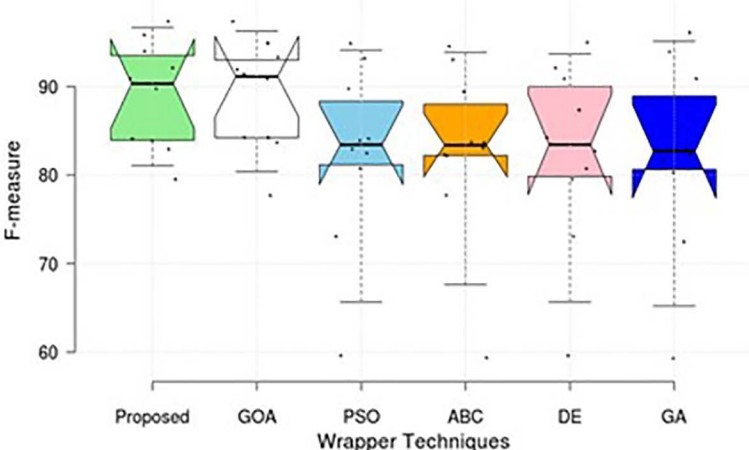

**Fig 3. Comparison of F-measure value for various wrapper techniques on BoT-IoT.**

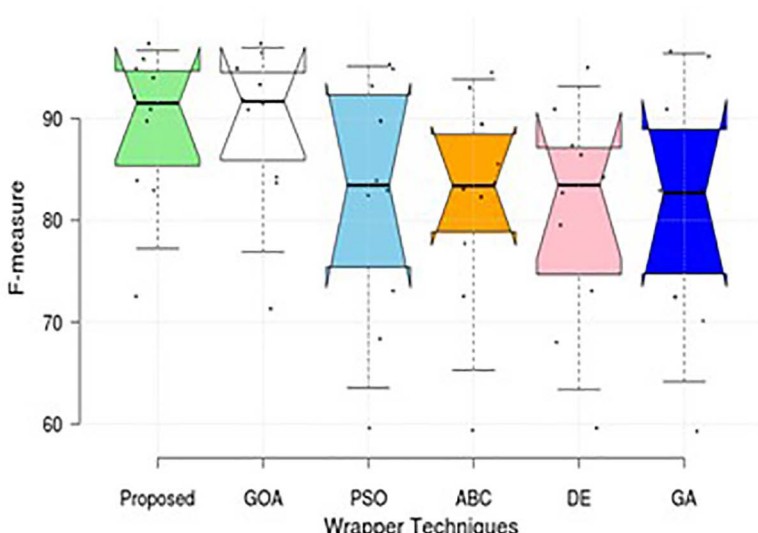

**Fig 4. Comparison of F-measure value for various wrapper techniques on UNSW-NB15.**

**Fig 5. Comparison of detection rate for various wrapper techniques on BoT-IoT.**

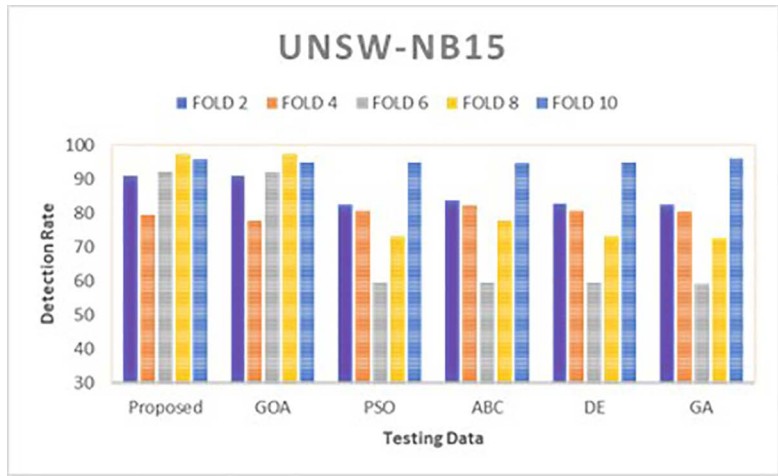

**Fig 6. Comparison of detection rate for various wrapper techniques on UNSW-NB15.**

testing datasets, as illustrated in Figs 5 and 6. Furthermore, Figs 7 and 8 demonstrates how the proposed method reduces the execution time required for testing in the UNSW-NB15 and BoT-IoT datasets.

The experimental results for each dataset are presented in Table 3, where the performance of the proposed method is compared against five different state-of-the-art procedures such as PSO, GOA, DE, ABC, and GA. Additionally, the results of testing our method on the BoT-IoT and UNSW-NB15 datasets clearly demonstrate that our strategy outperforms the feature subset in terms of accuracy, detection rate, and false positive rate. The evaluation measures in Table 4 demonstrate that, based on these data, our approach outperforms other cutting-edge approaches.

We evaluate the performance of the multi-class classification problem on Receiver Operating Characteristics (ROC) curve. This technique evaluate the performance of the proposed IDS framework and determine how well it addresses the categorisation problem. From Figs 9 and 10, we observed the relationship between the actual positive rate and the false positive rate for the classifiers used is the ROC curve. The 50% performance achieved by the used technique is

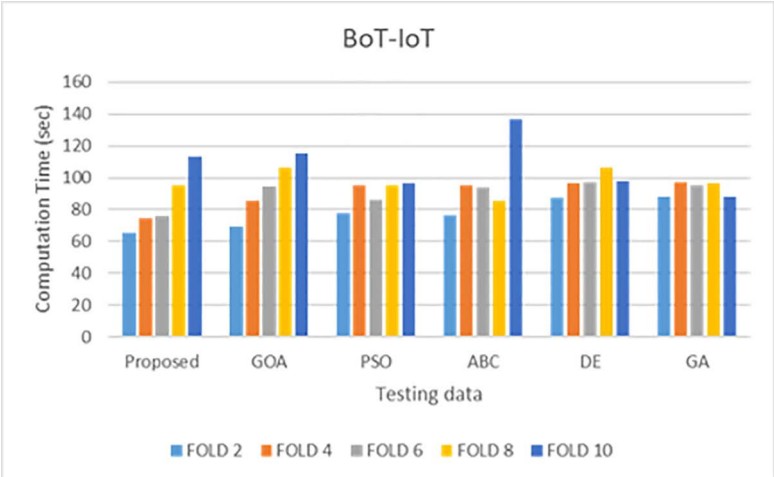

**Fig 7. Comparison of testing execution times for various wrapper techniques on the BoT-IoT.**

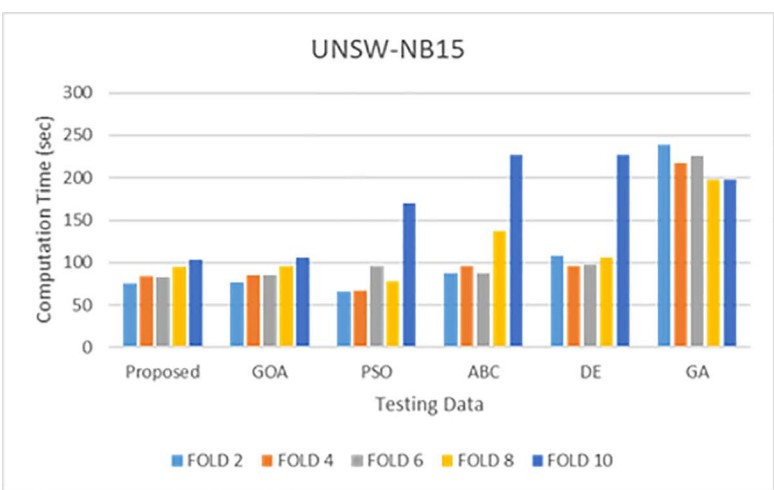

**Fig 8. Comparison of testing execution times for various wrapper techniques on the UNSW-NB15.**

indicated by the diagonal line of the ROC curve, which serves as a reference line. The approach 100% performance is displayed in the top-left corner of the ROC curve. Figs 9 and 10 shows the ROC curves of the tested models on the BoT-IoT and UNSW-NB15 datasets. One often-used metric for statistically evaluating various ROC curves is the Area Under the ROC Curve (AUROC). Compared to previous wrapper methodologies, Figs 9 and 10 demonstrates that our approach has an area under the curve value of 1, indicating that it is a suitable classifier for recognising attacks in the BoT-IoT and UNSW-NB15 datasets.

Table 4 illustrates that, compared to other current IDS strategies, such as [48,51], our approach achieves the highest detection accuracy. The proposed IDS technique outperforms other current state-of-the-art systems in terms of detection rate and achieves improved detection accuracy.

Table 3. Comparative analysis of the experimental performance, including all attacks of the BoT-IoT and UNSW-NB15 datasets.

| K-Fold | Measure | BoT-IoT | | | | | | UNSW-NB15 | | | | | |
|---|---|---|---|---|---|---|---|---|---|---|---|---|---|
| | | Proposed | GOA | PSO | ABC | DE | GA | Proposed | GOA | PSO | ABC | DE | GA |
| 2 | DR (%) | 93.99 | 93.34 | 93.18 | 93.03 | 90.9 | 90.89 | 90.9 | 90.85 | 82.46 | 83.71 | 82.69 | 82.39 |
| 4 | | 89.77 | 91.42 | 89.77 | 89.43 | 79.54 | 82.95 | 79.54 | 77.71 | 80.75 | 82.2 | 80.76 | 80.32 |
| 6 | | 84.09 | 84.25 | 84.09 | 83.73 | 92.1 | 93.92 | 92.1 | 91.93 | 59.61 | 59.37 | 59.61 | 59.29 |
| 8 | | 82.95 | 83.67 | 82.95 | 82.31 | 87.36 | 81.60 | 97.36 | 97.36 | 73.07 | 77.72 | 73.07 | 72.47 |
| 10 | | 83.89 | 84.23 | 83.91 | 83.07 | 84.21 | 83.03 | 95.82 | 94.9 | 94.89 | 94.53 | 95.01 | 96.11 |
| 2 | FPR (%) | 2.04 | 2.93 | 4.87 | 5.23 | 5.83 | 5.99 | 2.64 | 2.83 | 3.25 | 3.72 | 5.62 | 6.42 |
| 4 | | 2.42 | 2.63 | 2.74 | 3.13 | 4.96 | 5.93 | 2.99 | 3.02 | 3.83 | 3.91 | 2.94 | 4.67 |
| 6 | | 1.53 | 2.11 | 3.82 | 4.02 | 5.11 | 5.72 | 1.73 | 1.99 | 2.94 | 3.23 | 3.99 | 4.76 |
| 8 | | 0.89 | 1.35 | 2.76 | 4.73 | 4.99 | 7.23 | 1.89 | 2.89 | 3.69 | 4.88 | 4.68 | 6.61 |
| 10 | | 0.63 | 1.74 | 2.65 | 3.76 | 4.23 | 5.43 | 1.34 | 2.54 | 3.52 | 4.62 | 5.11 | 6.25 |
| 2 | Etr(sec) | 55.87 | 87.81 | 90.41 | 93.62 | 92.74 | 91.86 | 73.65 | 83.52 | 81.63 | 79.34 | 81.83 | 79.25 |
| 4 | | 61.53 | 63.82 | 64.81 | 62.91 | 59.52 | 58.99 | 67.93 | 101.34 | 87.62 | 88.92 | 91.82 | 89.03 |
| 6 | | 84.9 | 66.91 | 91.03 | 88.34 | 87.52 | 90.21 | 78.23 | 77.53 | 73.71 | 71.43 | 70.32 | 69.22 |
| 8 | | 72.53 | 71.35 | 68.34 | 72.53 | 68.01 | 70.12 | 81.32 | 79.34 | 81.73 | 77.02 | 74.52 | 77.82 |
| 10 | | 89.02 | 79.32 | 81.99 | 78.23 | 77.65 | 79.57 | 81.32 | 79.32 | 80.33 | 104.54 | 111.78 | 112.76 |
| 2 | Ete(sec) | 65.23 | 69.12 | 77.76 | 76.23 | 87.34 | 88.12 | 75.34 | 76.38 | 66.24 | 87.43 | 107.89 | 238.92 |
| 4 | | 74.65 | 85.72 | 95.14 | 95.44 | 96.48 | 96.88 | 84.02 | 84.99 | 67.52 | 96.21 | 96.67 | 217.61 |
| 6 | | 75.92 | 94.31 | 86.43 | 94.02 | 97.23 | 95.31 | 83.23 | 85.87 | 96.36 | 87.59 | 97.05 | 226 |
| 8 | | 94.9 | 106.43 | 95.31 | 85.54 | 106.43 | 96.62 | 94.77 | 96.2 | 77.47 | 136.92 | 106.24 | 197.64 |
| 10 | | 113.53 | 115.32 | 96.22 | 136.23 | 97.62 | 87.87 | 103.34 | 105.41 | 169.52 | 226.83 | 227.38 | 197.53 |

Table 4. Comparative analysis of the proposed in BoT-IoT and UNSW-NB15 datasets.

| Method | Dataset | DR | Accuracy | FPR |
|---|---|---|---|---|
| [48] | UNSW NB_15 | 97.00 | 97.17 | NA |
| [49] | UNSW NB_15 | NA | 91.02 | 0.2748 |
| [50] | UNSW NB_15 | NA | 95.82 | 1.53 |
| [51] | UNSW NB_15 | 99.20 | 99.90 | 0.603 |
| [52] | UNSW NB_15 | NA | 98.50 | NA |
| [53] | UNSW NB_15 | NA | 82.13 | NA |
| Proposed | UNSW NB_15 | 99.67 | 99.54 | 0.071 |
| [54] | BoT_IoT | NA | 99.60 | NA |
| [55] | BoT_IoT | 99.79 | 98.00 | NA |
| [29] | BoT_IoT | 88.37 | 88.37 | NA |
| [13] | BoT_IoT | NA | 98.20 | 1.281 |
| [56] | BoT_IoT | 95.00 | 95.00 | NA |
| [57] | BoT_IoT | 98.04 | NA | NA |
| Proposed | BoT_IoT | 99.91 | 99.45 | 0.075 |

Table 5 indicates that the ideal features selected by our technique are six and five, respectively, and it strongly identify the category of attacks in the IoT network. These qualities are essential parts of the IDS system. Table 5 presents the best-selected features and enumerates these attributes in the utilised datasets.

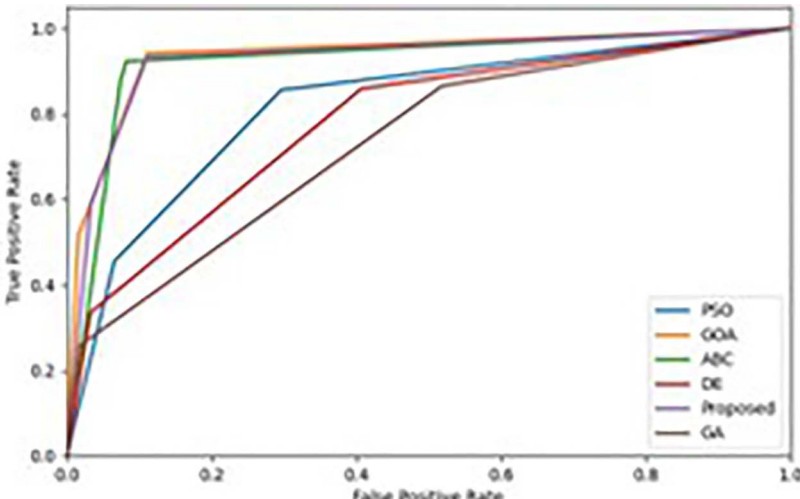

**Fig 9. ROC curves for wrapper techniques on the BoT-IoT.**

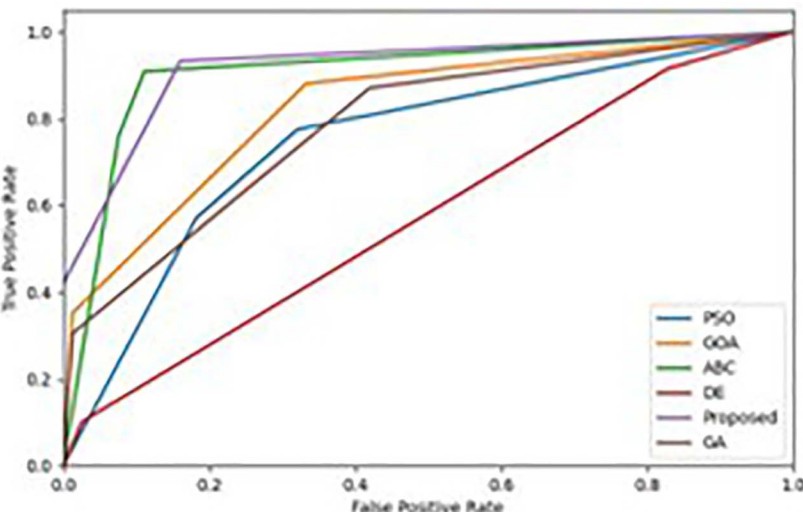

**Fig 10. ROC curves for wrapper techniques on the UNSW-NB15.**

**Table 5. Description of selected features.**

| Dataset | Attribute description |
| --- | --- |
| BoT-IoT | "Stime, Proto, state, srate, ltime, dbytes" |
| UNSW-NB15 | '' dsport, sloss, stcpb, smeansz, ct_ src_ dport_ ltm" |

The area under the Precision-Recall Curve (AUPR) values is used to quantify classifier performance. A perfect classifier receives an AUPR score of 1. The AUROC and AUPR values in Table 6 indicate that the proposed method outperforms other models in detecting network intrusions on all evaluated datasets. On the other hand, every strategy employed is quite well-suited to the situation.

**4.5.3. Statistical validation.** The statistical test achieves the relative performance evaluation between the two groups [58]. One of the most often used non-parametric statistical methods, the Friedman test [59], is used to rank algorithm performance. Identifying significant differences between algorithm outcomes. The null hypothesis assumes that there are no differences in the presentation of algorithms. The algorithm with the best performance is ranked lowest, while the algorithm with the worst performance is ranked highest. This study analyses evolutionary techniques and their differences using non-parametric test, which can be observed. These tests often lead to diverse results. To ensure sufficient reliability in our comparisons, we applied several nonparametric tests. The Friedman test is applied to perform multiple comparisons and check for the inequality amongst EA algorithms. Holm Hochberg, and Li procedures are used for pairwise performance evaluation. It is based on the null hypothesis that there is no dissimilarity in the presentation of algorithms. The best-performing algorithm gets the lowest rank, while the worst-performing algorithm receives the highest rank according to the following aspects:

1. The observed accuracy value for each algorithm and dataset pair.

2. For each IDS dataset, rank values from 2 (best result) to 5 (worst result).

3. For each method, average the ranks obtained in the IDS datasets to achieve the final rank.

The tests are applied to the accuracy obtained from IDS data sets. The average rank obtained by each algorithm on the IDS datasets is evaluated for responsibility according to Friedman's statistic, as shown in Table 7. Holm's procedure rejects those hypotheses that have an unadjusted p-value $\leq 0.01$. Li's procedure rejects those hypotheses that have an unadjusted p-value $\leq 0.022592$. Using Li and Holm's process, the post-hoc method for pairwise comparisons of metaheuristic methods further validates the suggested method. P-values that were determined by utilising post hoc techniques on the Friedman procedure findings are displayed in Table 8.

Therefore, the null hypothesis is rejected, indicating significant differences among the performances of the compared techniques. The proposed method is further validated by a post-hoc method for pairwise comparisons between EA methods, using Li and Holm's procedure. P-values were obtained by applying post-hoc methods to the results.

## 5. Conclusion

Indeed, this new era of cybersecurity has required a paradigm shift from modest defence mechanisms to sophisticated defence systems. To prevent integrity and malicious traffic identification, Internet of Things (IoT) network-based administrators are the cornerstone of cybersecurity. The faster development of IoT-based technology has permeated every aspect of our lives, including our bodies, homes, and living spaces, bringing various security concerns. A practicable intrusion

**Table 6. AUROC and AUPR values in BoT-IoT and UNSW-NB15 Datasets.**

| Datasets | Measures | GA | PSO | DE | ABC | GOA | CMGODE |
|---|---|---|---|---|---|---|---|
| BoT-IoT | AUROC | 97.89 | 96.73 | 94.75 | 95.15 | 95.40 | 98.89 |
| | AUPR | 96.32 | 95.95 | 95.12 | 94.75 | 96.11 | 98.91 |
| UNSW-NB15 | AUROC | 96.55 | 95.32 | 95.62 | 95.62 | 96.66 | 99.49 |
| | AUPR | 95.12 | 95.76 | 96.66 | 96.21 | 96.32 | 98.87 |

**Table 7. Summary of statistical results.**

| Algorithms | Ranking |
|---|---|
| Proposed | 2.125 |
| GOA | 2.875 |
| PSO | 4 |
| ABC | 3 |
| DE | 3.5 |
| GA | 5.5 |

**Table 8. Post Hoc Comparison Table for α = 0.05.**

| i | algorithm | z = (R0 − Ri)/SE | p | Holm Hochberg | Li |
|---|---|---|---|---|---|
| 5 | GA | 2.55126 | 0.010733 | 0.01 | 0.022592 |
| 4 | PSO | 1.417367 | 0.156376 | 0.0125 | 0.022592 |
| 3 | DE | 1.039402 | 0.298618 | 0.016667 | 0.022592 |
| 2 | ABC | 0.661438 | 0.508332 | 0.025 | 0.022592 |
| 1 | GOA | 0.566947 | 0.57075 | 0.05 | 0.05 |

detection technique, called chaotic multi-population GOA with DE (CMGODE), is proposed to address the second line of defence in protecting a network. The proposed CMGODE framework is designed to systematically identify an optimal subset of features that retain sufficient informational value for accurate intrusion detection, while operating under the critical constraints of an IoT environment. It incorporates chaotic multi-population and DE schemes into GOA to enhance performance, balance global and local search capabilities, and significantly increase population diversity. In the proposed approach, SVM serves as a fitness function that evaluates solutions based on specific performance criteria. The proposed technique performance is evaluated using two datasets, namely BoT-IoT and UNSW-NB15. In the multiclass classification of the BoT-IoT and UNSW-NB15 data, chaotic multi-population GOA obtains the best accuracy, detection rate, and low false positive rate compared to other state-of-the-art methods and popular wrapper techniques. The proposed method (CMGODE) achieved the highest detection rate (99.67% on UNSW-NB15, 99.91% on BoT-IoT) and the lowest false positive rate (0.071%, 0.075%) among state-of –the-arts, demonstrating superior overall reliability. While its accuracy remained highly competitive, it notably excelled in precision by significantly reducing false alarms across both IDS datasets. To further advance the capabilities of intelligent anomaly intrusion detection systems in IoT environments, future research can explore a hybrid manner that synergises the strengths of deep learning (DL) and evolutionary optimisation. Integrating deep learning and evolutionary optimisation approaches can overcome limitations in feature extraction, generalisation, and adaptability to dynamic network behaviours. Integrating our evolutionary algorithm with DL can automate and enhance the process by tuning hyper parameters and performing feature selection before input to DL models. This hybrid model can improve adaptability and generalisation to novel attack types, particularly in zero-day scenarios.

## Author contributions

**Conceptualization:** Shubhra Dwivedi, Alok Kumar Shukla.

**Data curation:** Shubhra Dwivedi, Alok Kumar Shukla, Diwakar Tripathi.

**Formal analysis:** Shubhra Dwivedi, Alok Kumar Shukla, Diwakar Tripathi.

**Funding acquisition:** Sunil Kumar Singh.

**Investigation:** Shubhra Dwivedi.

**Methodology:** Shubhra Dwivedi, Alok Kumar Shukla.

**Project administration:** Alok Kumar Shukla, Diwakar Tripathi.

**Resources:** Sunil Kumar Singh, Diwakar Tripathi.

**Software:** Shubhra Dwivedi, Diwakar Tripathi.

**Supervision:** Alok Kumar Shukla.

**Validation:** Sunil Kumar Singh, Diwakar Tripathi.

**Visualization:** Diwakar Tripathi.

**Writing – original draft:** Shubhra Dwivedi.

**Writing – review & editing:** Alok Kumar Shukla, Sunil Kumar Singh, Diwakar Tripathi.

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
