## [Decision Letter · Decision Letter 0]

21 Mar 2025

Dear Dr. Singh,

plosone@plos.org . . A rebuttal letter that responds to each point raised by the academic editor and reviewer(s). You should upload this letter as a separate file labeled 'Response to Reviewers'.A marked-up copy of your manuscript that highlights changes made to the original version. You should upload this as a separate file labeled 'Revised Manuscript with Track Changes'.An unmarked version of your revised paper without tracked changes. You should upload this as a separate file labeled 'Manuscript'.

We look forward to receiving your revised manuscript.

Kind regards,

Ayei Egu Ibor, PhD

Academic Editor

PLOS ONE

Journal Requirements:

4. Thank you for stating the following financial disclosure: [Manipal Institute of Technology Bengaluru, MAHE Bengaluru Campus].

5. Please provide a complete Data Availability Statement in the submission form, ensuring you include all necessary access information or a reason for why you are unable to make your data freely accessible. If your research concerns only data provided within your submission, please write "All data are in the manuscript and/or supporting information files" as your Data Availability Statement.

Reviewers' comments:

Reviewer's Responses to Questions

**Comments to the Author**

1. Is the manuscript technically sound, and do the data support the conclusions?

Reviewer #1: Yes

Reviewer #2: Yes

2. Has the statistical analysis been performed appropriately and rigorously?

Reviewer #1: Yes

Reviewer #2: I Don't Know

3. Have the authors made all data underlying the findings in their manuscript fully available?

Reviewer #1: Yes

Reviewer #2: No

4. Is the manuscript presented in an intelligible fashion and written in standard English?

Reviewer #1: Yes

Reviewer #2: Yes

Reviewer #1: Clarify the Novelty – Clearly articulate how CMGODE improves upon existing bio-inspired anomaly detection methods.

Expand Literature Review – Include a discussion on recent advancements in deep learning-based IDS for IoT security.

Dataset Preprocessing Details – Specify data cleaning, feature selection techniques, and normalization steps applied to BoT-IoT and UNSW-NB15 datasets.

Computational Efficiency – Provide runtime benchmarks on different hardware (e.g., IoT edge devices vs. high-performance servers).

Statistical Validation – Include statistical significance tests (e.g., t-test, ANOVA) to validate performance improvements.

Model Generalization – Discuss cross-network applicability and how the model performs under real-world traffic variations.

Real-Time Feasibility – Address latency concerns and resource consumption for real-time anomaly detection in IoT environments.

Comparison with DL-Based Methods – Benchmark performance against LSTM, CNN, and Transformer-based intrusion detection approaches.

Parameter Optimization Clarity – Provide insights on optimal parameter tuning for CMGODE and how it prevents local optima trapping.

Improve Figures & Tables – Ensure captions are self-explanatory, and unify redundant tables to enhance readability.

False Alarm Rate Analysis – Provide a deeper breakdown of False Positives and False Negatives in the detection results.

Adversarial Attack Resilience – Discuss the model’s robustness against adversarial threats and evasion attacks.

Future Work Directions – Expand on potential hybrid approaches (e.g., DL + Evolutionary Algorithms) for enhanced detection.

These refinements will enhance the clarity, impact, and applicability of the study.

Reviewer #2: The authors consider a modified bio-inspired algorithm called CMGODE anomaly intrusion detection system to handle sensitive information for effective operation. The proposed method modifies the fundamental GOA algorithm in three ways. Specifically, it avoids trapping in local minima and fails to consider exploration searchability. To address this, hybridizing the GOA algorithm with the Differential Evolution (DE) algorithm is integrated to enhance detection accuracy. Initially, the algorithm combines the concept of chaos to improve its exploitation abilities. After that, a multi-population strategy for variety and global search capabilities is integrated. Finally, the method incorporates DE into an improvement mechanism for solution quality.

However, the current version has the following shortcomings.

1) The chaotic initialization described in Eq. (12) and multi-population strategy are inadequately justified. Why was the logistic map chosen over other chaotic systems? How does the division into sub-populations prevent local optima?

2) Eq. (16) combines feature count, FPR, and TPR, but the weights for these terms are not specified. This raises concerns about bias toward feature reduction at the expense of detection accuracy.

3) The BoT-IoT and UNSW-NB15 datasets are widely used, but the authors employ only 10% of BoT-IoT for training. No justification is provided for this arbitrary subsetting, risking under representation of attack diversity.

4) The claimed superiority over GA, PSO, etc., lacks statistical validation. Tables 2--4 report average metrics without variance, making it impossible to assess robustness.

5) I suggest the authors should open their experiments in open platforms such as github, so that other researchers can compare their work with other ones.

6) While the proposed method achieves low FPR as shown in Table 4, the impact of such FPR in IoT contexts is not discussed.

7) Some variables such as \( ul_d, ll_d \) in Eq. (1) are not defined before their usage. Please check all variables throughout the whole paper.

8) Some fresh and important references closely related to the current work are missing. I think discussing the following works is beneficial for the readers.

- Yu S, Zhai R, Shen Y, Wu G, Zhang H, Yu S, et al. Deep Q-Network-based open-set intrusion detection solution for industrial Internet of Things. IEEE Internet of Things Journal. 2024;11(7):12536-50. http://doi.org/10.1109/JIOT.2023.3333903

- Shen S, Cai C, Li Z, Shen Y, Wu G, Yu S. Deep Q-Network-based heuristic intrusion detection against edge-based SIoT zero-day attacks. Applied Soft Computing. 2024; 150:111080.

- Yu S, Wang X, Shen Y, Wu G, Yu S, Shen S. Novel intrusion detection strategies with optimal hyper parameters for industrial Internet of Things based on stochastic games and double deep Q-networks. Ieee Internet of Things Journal. 2024;11(17):29132-45. http://doi.org/10.1109/JIOT.2024.3406386

- Shen S, Xie L, Zhang Y, Wu G, Zhang H, Yu S. Joint differential game and double deep Q-networks for suppressing malware spread in industrial Internet of Things. IEEE Transactions On Information Forensics and Security. 2023;18:5302-15. http://doi.org/10.1109/TIFS.2023.3307956

**Do you want your identity to be public for this peer review?** For information about this choice, including consent withdrawal, please see our For information about this choice, including consent withdrawal, please see our Privacy Policy .

Reviewer #1: No

Reviewer #2: No

While revising your submission, please upload your figure files to the Preflight Analysis and Conversion Engine (PACE) digital diagnostic tool, https://pacev2.apexcovantage.com/ . PACE helps ensure that figures meet PLOS requirements. To use PACE, you must first register as a user. Registration is free. Then, login and navigate to the UPLOAD tab, where you will find detailed instructions on how to use the tool. If you encounter any issues or have any questions when using PACE, please email PLOS at . PACE helps ensure that figures meet PLOS requirements. To use PACE, you must first register as a user. Registration is free. Then, login and navigate to the UPLOAD tab, where you will find detailed instructions on how to use the tool. If you encounter any issues or have any questions when using PACE, please email PLOS at figures@plos.org . Please note that Supporting Information files do not need this step.. Please note that Supporting Information files do not need this step.

---

## [Author Response · Author response to Decision Letter 1]

25 Apr 2025

Response to Reviewer Comments

We thank the Editor for handling the review process and the anonymous reviewers for their valuable comments and suggestions. In the revised manuscript, we have thoroughly addressed all the comments of the Editor, whose details are as follows.

Response to Reviewer 1

Comments to the Author: Clarify the Novelty – Clearly articulate how CMGODE improves upon existing bio-inspired anomaly detection methods.

Authors’ Response: We apologize for the lack of understandability and conciseness points related to Novelty in the previous manuscript. We have carefully revised the manuscript according to the reviewer's comments and added a novelty section, as seen in the introduction section 1.1, for better understanding.

Comments to the Author: Expand Literature Review – Include a discussion on recent advancements in deep learning-based IDS for IoT security.

Authors’ Response: We apologize for the lack of description of the Literature Reviews. In the revised manuscript, we have studied and compared the differences and shortcomings of different works based on recent advancements in deep learning-based IDS for IoT security.

Comments to the Author: Dataset Preprocessing Details – Specify data cleaning, feature selection techniques, and normalization steps applied to BoT-IoT and UNSW-NB15 datasets.

Authors’ Response: Thanks to the reviewer suggestion, in the revised manuscript, we have now provided preprocessing details based on data cleaning, feature selection techniques, and normalization steps applied to both datasets.

Comments to the Author: Computational Efficiency – Provide runtime benchmarks on different hardware (e.g., IoT edge devices vs. high-performance servers).

Authors’ Response: Sincere thanks for your suggestion and comment on our manuscript. Several key factors, including population size, dimensionality, chaos-inspired developments, multi-population strategies, and Differential Evolution, influence the computational complexity of Chaotic Multi-population Grasshopper Optimization Algorithm (CMGODE). Understanding the complexity of CMGODE requires breaking down its significant computational steps and analyzing their impact on overall efficiency.

Assume that D is the dimension of individuals, tm is the number of iterations, and pop is the number of solutions. The three primary steps of the conventional Grasshopper Optimisation Algorithm (GOA) are iterative position updating, fitness evaluation, and population initialisation. The initialisation stage has an O(pop*D) complexity since the population of pop grasshoppers must be set up in D-dimensional space. The fitness evaluation procedure's complexity, which depends on the computation training function complexity £, is O(pop*£). O(pop*D) operations are needed for each iteration of the grasshoppers' attraction-repulsion-based iterative movement. The total complexity of a typical GOA is O(tm*pop*D), since GOA executes for a maximum of tm iterations.

The computing process is impacted by adding chaos-inspired changes to CMGODE, especially in search tactics and position updating. The CMGODE method uses a population initialisation strategy inspired by chaos to increase the diversity of initial solutions and prevent premature convergence. Chaotic systems show deterministic but pseudo-random behaviour that guarantees uniform search space coverage, unlike traditional random initialisation, which may result in clusters of closely spaced solutions; the complexity stays at O(pop*D). Even though this step adds more probability calculations, the asymptotic complexity stays at O(pop*D). Consequently, the asymptotic complexity of the chaos modifications in CMGODE is O(tm*pop*D), the same as that of GOA.

By segmenting the population into subP subpopulations, the multi-population technique in CMGODE further improves global searchability. Every subpopulation develops independently, with people migrating periodically to increase diversity. The complexity of the division step is O(Pop), whereas the evolution of independent subpopulations is O(t* m*pop/subP*D). The regular subpopulation movement adds a small O(pop) overhead. Pop is usually modest. Therefore, O(tm*pop*D) continues to be the dominant complexity term.

CMGOA incorporates Differential Evolution (DE) to enhance solution quality, which entails crossover, mutation, and selection procedures. O(pop*D) calculations are needed for the mutation step, which involves transforming three randomly chosen solutions using vectors. Lastly, O(pop) comparisons are required for the selection stage, which keeps the best solutions. The overall order stays the same because these operations are carried out at each iteration, meaning that the total extra complexity from DE stays O(tm*pop*D). Thus, the final computational complexity of CMGODE, which may be written as O[(tm*pop*D)+O(tm*pop)+(pop∗D+£)], remains after the computational steps provided by chaos operations, multi-population strategies, and DE improvements. Therefore, our method has low computational complexity compared to traditional evolutionary methods.

Comments to the Author: Statistical Validation – Include statistical significance tests (e.g., t-test, ANOVA) to validate performance improvements.

Authors’ Response: We appreciate this query. The statistical test achieves the relative performance evaluation between the two groups of members [1]. In this study, evolutionary techniques and their differences are analysed using a non-parametric test, which can be observed. These tests often lead to diverse results. To ensure sufficient reliability in our comparisons, we applied several nonparametric tests. Friedman test [2] is applied to perform multiple comparisons and check for the inequality amongst EA algorithms. Holm Hochberg and Li [3][4] procedures are used for pairwise performance evaluation.

It is based on the null hypothesis that there is no dissimilarity in the presentation of algorithms. The best performing algorithm gets the lowest rank while the worst performing algorithm receives the highest rank (see Tables 7 and 8) in the revised manuscript) according to the following aspects:

1. The observed detection accuracy value for each algorithm and dataset pair.

2. For each IDS dataset, rank values from 2 (best result) to 5 (worst result).

3. For each method, average the ranks obtained in the IDS datasets to achieve the final rank.

The tests are applied to the accuracy obtained from IDS data sets. The average rank obtained by each algorithm on IDS datasets is evaluated for responsibility according to Friedman’s statistic, which is shown in the Table 7. The proposed method is further validated by the post-hoc method for pairwise comparisons between EA methods using Li and Holm’s procedure. P-values obtained by applying post hoc methods over the results of the Friedman procedure as shown in Table 8.

[1]. Iantovics, L. B., Dehmer, M., & Emmert-Streib, F. (2018). MetrIntSimil—An accurate and robust metric for comparison of similarity in intelligence of any number of cooperative multiagent systems. Symmetry, 10(2), 48.

[2]. García, Salvador, Alberto Fernández, Julián Luengo, and Francisco Herrera. "Advanced nonparametric tests for multiple comparisons in the design of experiments in computational intelligence and data mining: Experimental analysis of power." Information Sciences 180, no. 10 (2010): 2044-2064.

[3]. Benavoli, Alessio, Giorgio Corani, and Francesca Mangili. "Should we really use post-hoc tests based on mean-ranks?." The Journal of Machine Learning Research 17, no. 1 (2016): 152-161.

[4]. Hochberg, Yosef. "A sharper Bonferroni procedure for multiple tests of significance." Biometrika 75, no. 4 (1988): 800-802.

Comments to the Author: Model Generalization – Discuss cross-network applicability and how the model performs under real-world traffic variations.

Authors’ Response: Sincere thanks for your suggestion and comment on our manuscript. In the revised manuscript, we have added cross-network applicability, and the model performs under real-world traffic variations.

Comments to the Author: Real-Time Feasibility – Address latency concerns and resource consumption for real-time anomaly detection in IoT environments.

Authors’ Response: The experiment has been conducted on two datasets, which may not constrain advanced comparative methods and the corresponding experimental analysis. In the revised manuscript, we discussed comparative methods and experimental analysis more. This experiment may capture the full variability and complexity inherent in IDS data by relying on several datasets. This could potentially impact the robustness and generalizability of the findings, as the results are derived from a narrower scope of data. Furthermore, the datasets used in the experiment may have provided extensive insight into the study. A more thorough investigation involving a wider range of investigations would provide a more comprehensive understanding of the problem and strengthen the conclusions, which would help ensure they are more widely applicable.

Comments to the Author: Comparison with DL-Based Methods – Benchmark performance against LSTM, CNN, and Transformer-based intrusion detection approaches.

Author’s Response: We apologize for the lack of comparison with DL-Based Methods. In the revised manuscript, we compared DL-based methods based on LSTM, CNN, and Transformer-based intrusion detection approaches. Furthermore, we added research related to DL-based models in our work.

Comments to the Author: Parameter Optimization Clarity – Provide insights on optimal parameter tuning for CMGODE and how it prevents local optima trapping.

Author’s Response: The effectiveness of our algorithm significantly depends on the optimal configuration of its control parameters, which directly influence its convergence behavior and solution quality. Key parameters include the number of iterations, population size, chaotic map selection, sub-populations, crossover and mutation rates from the DE, and the frequency of elite migration among sub-populations. Through extensive empirical analysis, parameter settings yield consistent and optimal performance across both datasets: population size, maximum iterations, three sub-populations, and DE parameters such as a mutation factor and crossover rate. The Logistic chaotic map was also selected for its superior performance in enhancing exploration diversity.

To address the common issue of premature convergence and local optima trapping, CMGODE integrates multiple mechanisms. First, the chaotic map initialization introduces nonlinear and unpredictable population distributions, allowing the search process to escape from deceptive basins of attraction. Second, the multi-population structure ensures parallel search space exploration, reducing the chances of the entire population stagnating around sub-optimal solutions. Periodic elite individual exchange among sub-populations further accelerates convergence without sacrificing diversity. Finally, the Differential Evolution refinement phase intensifies exploitation around the best solutions, providing fine-grained tuning and preventing the algorithm from settling on locally optimal but globally inferior regions. These strategies ensure robust parameter tuning and effective optimization, leading to high detection accuracy and model generalization.

Comments to the Author: Improve Figures & Tables – Ensure captions are self-explanatory, and unify redundant tables to enhance readability.

Author’s Response: We apologize for the lack of presentation of Figures & Tables in the previous version of the manuscript. We tried to improve the quality of the manuscript, but we did not meet your expectations. We have restructured Figures & Tables per the reviewer's suggestion and enhanced the quality and readability of Figures & Tables. We hope that this revised manuscript meets your expectations.

Comments to the Author: False Alarm Rate Analysis – Provide a deeper breakdown of False Positives and False Negatives in the detection results.

Author’s Response: Sincere thanks for your suggestion and comment on our manuscript. In the revised manuscript, we have provided a deeper breakdown of the experimental results to fully demonstrate the applicability and robustness of the proposed approach in different scenarios in terms of False Alarm Rate Analysis.

Comments to the Author: Adversarial Attack Resilience – Discuss the model’s robustness against adversarial threats and evasion attacks.

Author’s Response: Sincere thanks for your suggestion and comment on our manuscript. In the revised manuscript, we have discussed the model's robustness against adversarial threats and evasion attacks.

Comments to the Author: Future Work Directions – Expand on potential hybrid approaches (e.g., DL + Evolutionary Algorithms) for enhanced detection. These refinements will enhance the clarity, impact, and applicability of the study.

Author’s Response: Sincere thanks for your suggestion and comment on our manuscript. In the revised manuscript, we have added future work directions on potential hybrid approaches for enhanced detection.

Response to Reviewer 2

Comments to the Author: The authors consider a modified bio-inspired algorithm called CMGODE anomaly intrusion detection system to handle sensitive information for effective operation. The proposed method modifies the fundamental GOA algorithm in three ways. Specifically, it avoids trapping in local minima and fails to consider exploration searchability. To address this, hybridizing the GOA algorithm with the Differential Evolution (DE) algorithm is integrated to enhance detection accuracy. Initially, the algorithm combines the concept of chaos to improve its exploitation abilities. After that, a multi-population strategy for variety and global search capabilities is integrated. Finally, the method incorporates DE into an improvement mechanism for solution quality.

However, the current version has the following shortcomings.

1) The chaotic initialization described in Eq. (12) and multi-population strategy are inadequately justified. Why was the logistic map chosen over other chaotic systems? How does the division into sub-populations prevent local optima?

Author’s Response: We apologize for the lack of understanding and justification related to chaotic and multi-population strategies in the previous version of the manuscript. We have carefully revised the manuscript according to the reviewer comments and for convenience here also.

The logistic map was chosen due to its simplicity, computational efficiency, and well-known chaotic behavior, which is widely validated in EA optimization literature [5][6]. Its capability to generate diverse and non-repetitive initial solutions helps avoid premature convergence. Furthermore, compared to other chaotic systems [7], the logistic map demonstrates a good balance between randomness and control. It is particularly suitable for enhancing the exploration-exploitation trade-off in the early stages of population initialization. Nevertheless, we acknowledge that other chaotic maps may offer alternative benefits, and a comparative analysis could be a promising extension for future work.

The division into sub-populations is designed to enhance the diversity of the search space and reduce the risk of stagnation in local optima. Each sub-population independently explores different search space regions, increasing our algorithm's global search capability. Intermittent communication or information exchange among sub-populations enables the algorithm to share promising solutions, thus balancing convergence and diversity. Empirical results in our ablation studies demonstrated that the multi-population mechanism significantly contributes to avoiding local optima and improves overall detection accuracy and robustness.

5.Syamarao, Sowmya Karanam, Amith Shekhar Chandrashekhar, Anupkumar Manohara Bongale, and Deepak Sudhakar Dharrao. "Network Intrusion Detection System in Internet of Things Using Chaotic Elite Guidance Learning Strategy-based Lotus Effect Optimization Algorithm." International Journal of Intelligent Engineering & Systems 18, no. 1 (2025).

6.Bamakan, Seyed Mojtaba Hosseini, Huadong Wang, Tian Y

---

## [Decision Letter · Decision Letter 1]

12 Jan 2026

Dear Dr. Singh,

plosone@plos.org . . A letter that responds to each point raised by the academic editor and reviewer(s). You should upload this letter as a separate file labeled 'Response to Reviewers'.A marked-up copy of your manuscript that highlights changes made to the original version. You should upload this as a separate file labeled 'Revised Manuscript with Track Changes'.An unmarked version of your revised paper without tracked changes. You should upload this as a separate file labeled 'Manuscript'.

We look forward to receiving your revised manuscript.

Kind regards,

Vincent Omollo Nyangaresi, Ph.D

Academic Editor

PLOS One

Journal Requirements:

Reviewers' comments:

Reviewer's Responses to Questions

**Comments to the Author**

Reviewer #1: All comments have been addressed

Reviewer #2: All comments have been addressed

2. Is the manuscript technically sound, and do the data support the conclusions?

Reviewer #1: Yes

Reviewer #2: Yes

3. Has the statistical analysis been performed appropriately and rigorously?

Reviewer #1: Yes

Reviewer #2: Yes

4. Have the authors made all data underlying the findings in their manuscript fully available?

Reviewer #1: Yes

Reviewer #2: (No Response)

5. Is the manuscript presented in an intelligible fashion and written in standard English?

Reviewer #1: Yes

Reviewer #2: (No Response)

Reviewer #1: The revised manuscript shows clear and commendable improvement over the original submission. The authors have systematically addressed all prior reviewer comments with well-documented changes:

A new novelty section was added (Section 1.1), effectively highlighting the contribution of CMGODE.

The literature review now includes recent work on deep learning-based IDS systems.

The methodology and algorithm (chaotic multi-population GOA with DE) are thoroughly explained with mathematical detail, parameter settings, and motivation for design choices (e.g., use of the logistic map, sub-populations, DE steps).

Statistical tests (Friedman test, Holm, and Li post hoc tests) are included and interpreted properly.

The manuscript also now discusses model generalization, real-time feasibility, adversarial robustness, and false alarm rates — all critical for intrusion detection in IoT environments.

Response tables and revised figures have been improved in structure and clarity.

Minor issues remain only at the language level. A light editorial review for grammar and phrasing will enhance the manuscript’s clarity. Additionally, please add explicit links to the public datasets (BoT-IoT and UNSW-NB15) in the Data Availability Statement for full transparency.

Overall, this is a strong and timely contribution to IoT anomaly detection and metaheuristic optimization research. I recommend acceptance with minor revisions (language and dataset URL clarity).

Reviewer #2: (No Response)

**Do you want your identity to be public for this peer review?** For information about this choice, including consent withdrawal, please see our For information about this choice, including consent withdrawal, please see our Privacy Policy .

Reviewer #1: No

Reviewer #2: No

---

## [Author Response · Author response to Decision Letter 2]

16 Jan 2026

Response to Reviewer 1

Comments to the Author: The revised manuscript shows clear and commendable improvement over the original submission. The authors have systematically addressed all prior reviewer comments with well-documented changes:

A new novelty section was added (Section 1.1), effectively highlighting the contribution of CMGODE.

The literature review now includes recent work on deep learning-based IDS systems.

The methodology and algorithm (chaotic multi-population GOA with DE) are thoroughly explained with mathematical detail, parameter settings, and motivation for design choices (e.g., use of the logistic map, sub-populations, DE steps).

Statistical tests (Friedman test, Holm, and Li post hoc tests) are included and interpreted properly.

The manuscript also now discusses model generalization, real-time feasibility, adversarial robustness, and false alarm rates — all critical for intrusion detection in IoT environments.

Response tables and revised figures have been improved in structure and clarity.

Authors’ Response: We are thankful to the reviewer for considering all comments and recommended to acceptance with minor revisions.

Comments to the Author: Minor issues remain only at the language level. A light editorial review for grammar and phrasing will enhance the manuscript’s clarity.

Author’s Response: Sincerely thankful for accepting our manuscript. We apologize for the lack of readability of the manuscript; in the previous version of the manuscript, we tried to improve the linguistic quality of the manuscript, but we did not meet your expectations. We have rechecked the revised manuscript by a native English speaker from the abstract to the conclusion section. Several minor grammatical and spelling edits have been made throughout the text that have improved linguistic quality as well as checked the numbering in references. We hope that this revised manuscript meets your expectations.

Comments to the Author: Additionally, please add explicit links to the public datasets (BoT-IoT and UNSW-NB15) in the Data Availability Statement for full transparency.

Author Response: We apologize for the lack of explicit links of the public datasets (BoT-IoT and UNSW-NB15) in previous manuscript. In response, we have included links of the public datasets (BoT-IoT and UNSW-NB15) in the Data Availability Statement.

---

## [Decision Letter · Decision Letter 2]

1 Feb 2026

Dear Dr. Singh,

plosone@plos.org . . A letter that responds to each point raised by the academic editor and reviewer(s). You should upload this letter as a separate file labeled 'Response to Reviewers'.A marked-up copy of your manuscript that highlights changes made to the original version. You should upload this as a separate file labeled 'Revised Manuscript with Track Changes'.An unmarked version of your revised paper without tracked changes. You should upload this as a separate file labeled 'Manuscript'.

We look forward to receiving your revised manuscript.

Kind regards,

Vincent Omollo Nyangaresi, Ph.D

Academic Editor

PLOS One

Journal Requirements:

Reviewers' comments:

Reviewer's Responses to Questions

**Comments to the Author**

Reviewer #1: All comments have been addressed

2. Is the manuscript technically sound, and do the data support the conclusions?

Reviewer #1: Yes

3. Has the statistical analysis been performed appropriately and rigorously?

Reviewer #1: Yes

4. Have the authors made all data underlying the findings in their manuscript fully available?

Reviewer #1: Yes

5. Is the manuscript presented in an intelligible fashion and written in standard English?

Reviewer #1: Yes

Reviewer #1: The revised manuscript demonstrates a substantial and commendable improvement over the original submission. The authors have carefully addressed prior reviewer concerns and significantly strengthened the manuscript in several key areas:

Novelty: Section 1.1 now clearly articulates the novelty of the CMGODE algorithm, justifying the use of chaos theory and multi-population strategies in the context of intrusion detection.

Methodology: The proposed algorithm is thoroughly explained with mathematical formulations, flowcharts (see Figures 1 and 2, pages 16 and 22), and pseudocode, enhancing reproducibility.

Experimental Rigor: Use of BoT-IoT and UNSW-NB15 datasets, comprehensive pre-processing, and comparison against multiple algorithms strengthen the empirical foundation.

Statistical Validation: Inclusion of statistical hypothesis testing (e.g., Friedman test) supports the significance of the findings.

Practical Considerations: Discussion on generalization, adversarial robustness, false alarm rates, and real-time feasibility is a major strength.

Recommendations:

Ensure the manuscript undergoes final copyediting for minor language issues.

Consider summarizing key experimental findings in a visual table in the conclusion section for quicker reference by readers.

Once these minor editorial issues are addressed, I believe this manuscript is suitable for publication.

**Do you want your identity to be public for this peer review?** For information about this choice, including consent withdrawal, please see our For information about this choice, including consent withdrawal, please see our Privacy Policy .

Reviewer #1: No

---

## [Author Response · Author response to Decision Letter 3]

8 Feb 2026

Response to Reviewer 1

Comments to the Author: The revised manuscript demonstrates a substantial and commendable improvement over the original submission. The authors have carefully addressed prior reviewer concerns and significantly strengthened the manuscript in several key areas:

Novelty: Section 1.1 now clearly articulates the novelty of the CMGODE algorithm, justifying the use of chaos theory and multi-population strategies in the context of intrusion detection.

Methodology: The proposed algorithm is thoroughly explained with mathematical formulations, flowcharts (see Figures 1 and 2, pages 16 and 22), and pseudocode, enhancing reproducibility. Experimental Rigor: Use of BoT-IoT and UNSW-NB15 datasets, comprehensive pre-processing, and comparison against multiple algorithms strengthen the empirical foundation. Statistical Validation: Inclusion of statistical hypothesis testing (e.g., Friedman test) supports the significance of the findings. Practical Considerations: Discussion on generalization, adversarial robustness, false alarm rates, and real-time feasibility is a major strength.

Authors’ Response: We are thankful to the reviewer for considering all comments and recommended to acceptance with minor revisions.

Comments to the Author: Ensure the manuscript undergoes final copyediting for minor language issues. Consider summarizing key experimental findings in a visual table in the conclusion section for quicker reference by readers.

Once these minor editorial issues are addressed, I believe this manuscript is suitable for publication.

Author’s Response: Sincerely thankful for accepting our manuscript. We apologize for the lack of readability of the manuscript; in the previous version of the manuscript, we tried to improve the linguistic quality of the manuscript, but we did not meet your expectations. We have rechecked the revised manuscript by a native English speaker from the abstract to the conclusion section. Several minor grammatical and spelling edits have been made throughout the text that have improved linguistic quality, as well as checked the numbering in references. Furthermore, in the revised manuscript, we have summarized key experimental findings in the conclusion section for better understanding.

We hope that this revised manuscript meets your expectations.

---

## [Editor Report · Decision Letter 3]

25 Feb 2026

Safeguarding Against External Intrusions Utilizing Adaptive Bio-Inspired Multi-Population Anomaly Detection for IoT Network

PONE-D-25-05218R3

Dear Dr. Singh,

We’re pleased to inform you that your manuscript has been judged scientifically suitable for publication and will be formally accepted for publication once it meets all outstanding technical requirements.

Kind regards,

Vincent Omollo Nyangaresi, Ph.D

Academic Editor

PLOS One